# Forkhead transcription factor FKH-8 cooperates with RFX in the direct regulation of sensory cilia in *Caenorhabditis elegans*

Rebeca Brocal-Ruiz[1], Ainara Esteve-Serrano[1], Carlos Mora-Martínez[1], Maria Luisa Franco-Rivadeneira[2], Peter Swoboda[3], Juan J Tena[4], Marçal Vilar[2], Nuria Flames[1]*

[1]Developmental Neurobiology Unit, Instituto de Biomedicina de Valencia IBV-CSIC, Valencia, Spain; [2]Molecular Basis of Neurodegeneration Unit, Instituto de Biomedicina de Valencia IBV-CSIC, Valencia, Spain; [3]Department of Biosciences and Nutrition. Karolinska Institute. Campus Flemingsberg, Stockholm, Sweden; [4]Centro Andaluz de Biología del Desarrollo (CABD), Consejo Superior de Investigaciones Científicas/Universidad Pablo de Olavide, Seville, Spain

**\*For correspondence:**
nflames@ibv.csic.es

**Competing interest:** The authors declare that no competing interests exist.

**Abstract** Cilia, either motile or non-motile (a.k.a primary or sensory), are complex evolutionarily conserved eukaryotic structures composed of hundreds of proteins required for their assembly, structure and function that are collectively known as the ciliome. Ciliome gene mutations underlie a group of pleiotropic genetic diseases known as ciliopathies. Proper cilium function requires the tight coregulation of ciliome gene transcription, which is only fragmentarily understood. RFX transcription factors (TF) have an evolutionarily conserved role in the direct activation of ciliome genes both in motile and non-motile cilia cell-types. In vertebrates, FoxJ1 and FoxN4 Forkhead (FKH) TFs work with RFX in the direct activation of ciliome genes, exclusively in motile cilia cell-types. No additional TFs have been described to act together with RFX in primary cilia cell-types in any organism. Here we describe FKH-8, a FKH TF, as a direct regulator of the sensory ciliome genes in *Caenorhabditis elegans*. FKH-8 is expressed in all ciliated neurons in C. *elegans*, binds the regulatory regions of ciliome genes, regulates ciliome gene expression, cilium morphology and a wide range of behaviors mediated by sensory ciliated neurons. FKH-8 and DAF-19 (*C. elegans* RFX) physically interact and synergistically regulate ciliome gene expression. *C. elegans* FKH-8 function can be replaced by mouse FOXJ1 and FOXN4 but not by other members of other mouse FKH subfamilies. In conclusion, RFX and FKH TF families act jointly as direct regulators of ciliome genes also in sensory ciliated cell types suggesting that this regulatory logic could be an ancient trait predating functional cilia sub-specialization.

## Editor's evaluation

Sensory cilia are highly specialized organelles whose development and function requires complex machinery. In this important work, the authors use a convincing set of genetic, bioinformatic, and biochemical approaches in *C. elegans* to demonstrate that the forkhead transcription factor FKH-8 acts cooperatively with the RFX factor DAF-19 to activate the expression of many cilium genes. These findings indicate that forkhead factors have an ancient, conserved role in the development of sensory cilia, making this paper of interest to a variety of developmental and cell biologists.

## Introduction

Eukaryotic cilia are complex and highly organized organelles defined as specialized membrane protrusions formed from a stereotyped assembly of microtubules. Cilia are composed of hundreds of proteins, required for their assembly, structure and function, which are collectively known as the ciliome (*Figure 1A*). Cilia can be classified into motile or non-motile based on their function and structure: motile cilia are responsible for propelling cells or generating fluid flow while non-motile (a.k.a primary or sensory) cilia function as cellular antennae to sense extracellular stimuli (*Choksi et al., 2014*). Cilia appeared early in eukaryotic evolution and it is thought that in ancient unicellular eukaryotes cilia displayed mixed motile and sensory functions (*Mitchell, 2017*). In multicellular invertebrates, primary and motor cilia are restricted to specific cell types. In contrast, in vertebrates, primary cilia are present almost in every cell, including neurons, while motile cilia are present only in specialized cell types.

Most ciliome components are shared between motile and primary cilia and are referred as 'core' ciliome (*Figure 1A*). In addition, motile cilia usually contain specialized axonemal dyneins, other motile-specific components and specific signaling proteins while the membrane of sensory cilia is decorated with receptors that trigger downstream signaling cascades when they are activated by small molecules, mechanical perturbations, or radiation.

The importance and wide range of cilia functions are underscored by the large number of congenital disorders caused by mutations in genes coding for ciliome components, which are collectively called ciliopathies (*Andreu-Cervera et al., 2021*; *Horani and Ferkol, 2021*; *Lucas et al., 2020*; *Tobin and Beales, 2009*). These disorders cause a broad spectrum of symptoms including retinal degeneration, polycystic kidney, deafness, polydactyly, brain and skeletal malformations, infertility, morbid obesity and mental retardation. Importantly, there are still many 'orphan ciliopathies', which correspond to congenital disorders classified as ciliopathies by phenotype but with yet unidentified causal mutations. Genetic variants lying in coding genes (including mutations in the ciliome genes) are easier to identify as causal mutations, however, most variants associated to human diseases lie in the non-coding genome (*Chatterjee and Ahituv, 2017*; *Battle et al., 2017*; *Timpson et al., 2018*). It is currently thought that some of these non-coding variants act as regulatory mutations affecting gene expression. Thus, regulatory mutations affecting ciliome gene expression might underlie many orphan ciliopathies. Understanding the molecular mechanisms that ensure correct co-regulation of ciliome genes is then of utmost importance.

Little is known about the direct transcriptional co-regulation of ciliome gene expression (*Choksi et al., 2014*; *Lewis and Stracker, 2021*; *Thomas et al., 2010*). In 2000, pioneering work in *Caenorhabditis elegans* identified DAF-19, an RFX family transcription factor (TF), as a direct regulator of ciliome gene expression in the ciliated sensory neurons (*Swoboda et al., 2000*). This work was followed by numerous reports on the role of different members of the RFX TF family as direct ciliome gene regulators both in primary and motile cilia cell-types in several animal models including *Drosophila melanogaster*, *Danio rerio*, *Xenopus laevis,* and *Mus musculus* (*Ashique et al., 2009*; *Bonnafe et al., 2004*; *Chung et al., 2012*; *Dubruille et al., 2002*; *Liu et al., 2007*) and in humans (*Sugiaman-Trapman et al., 2018*). FOXJ1, an ancient member of the Forkhead family, also acts as a direct activator of ciliome gene transcription in several vertebrates, but its role is limited to cell types containing motile cilia (*Brody et al., 2000*; *Chen et al., 1998*; *Stubbs et al., 2008*; *Vij et al., 2012*; *Yu et al., 2008*). Thus, currently additional TFs acting together with RFX TFs in the direct regulation of the ciliome gene expression in sensory cilia cell-types are unknown in any organism.

Here, we take advantage of the amenability of *C. elegans* for genetic studies to understand the transcriptional regulatory logic of the non-motile primary cilliome genes. *C. elegans* contains sensory but not motile cilia. In hermaphrodites, sensory cilia are found in 25 out of the 118 neuronal types known as the ciliated sensory system (*Inglis et al., 2007*; *Figure 1B*). We find that FKH-8, a FKH TF, is expressed in all ciliated sensory neurons in *C. elegans*, with an onset of expression concomitant to the start of ciliome gene expression. Chromatin immuno-precipitation and sequencing (ChIP-seq) data analysis shows that FKH-8 binds to a broad range of ciliome genes, at locations often near X-box motifs, the binding sites for DAF-19/RFX. *fkh-8* mutants show decreased ciliome reporter gene expression, cilia morphology abnormalities and deficits in a wide range of behaviors mediated by sensory cilia. In addition, we find FKH-8/FKH and DAF-19/RFX can physically interact and act synergistically in the regulation of ciliome genes. Finally, we show that mouse FoxJ1 and FoxN4, two ancient FKH TFs

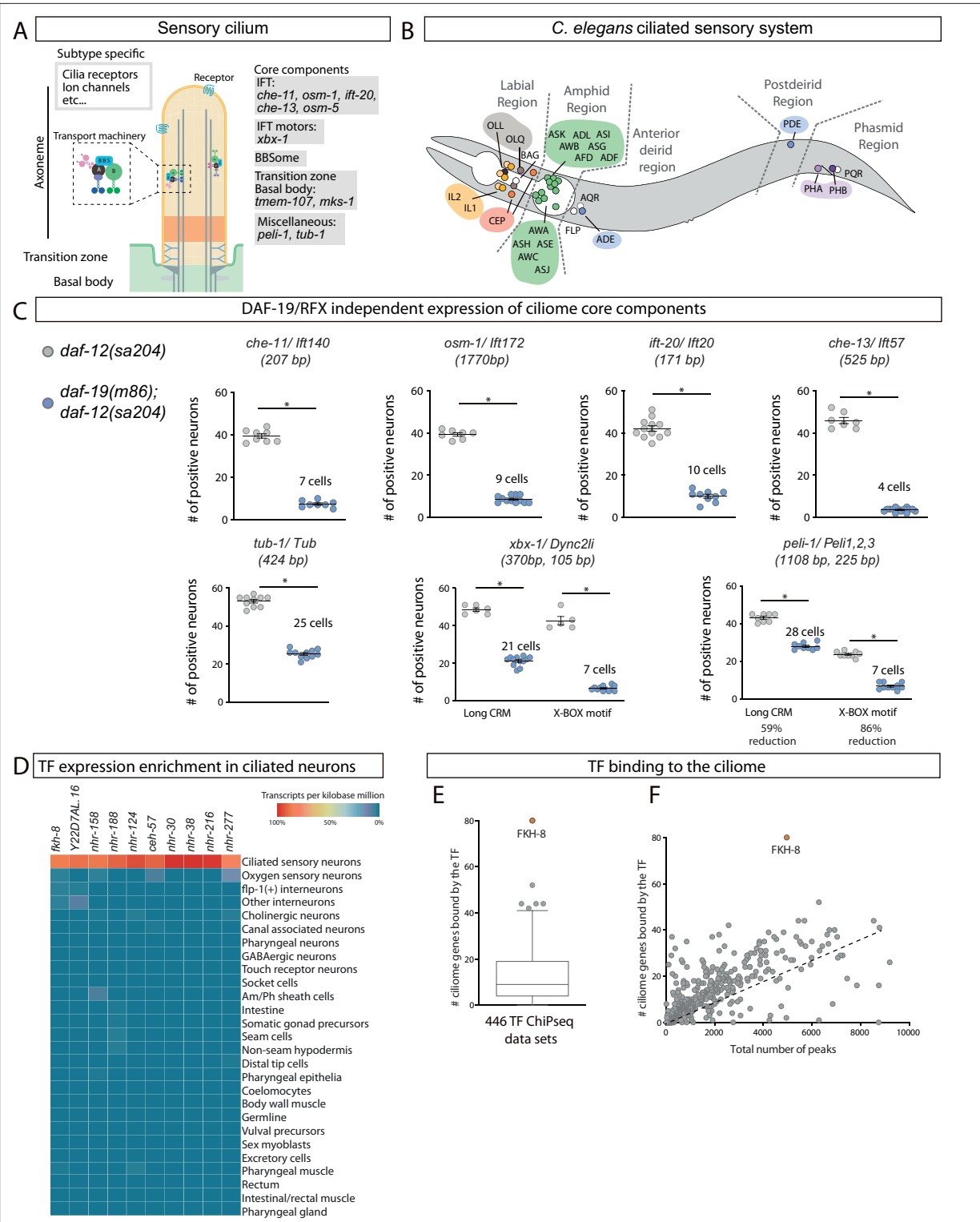

**Figure 1.** FKH-8 is a candidate for direct regulation of ciliome gene expression in *C. elegans*. (**A**) Schema for a sensory cilium. Cilia components (ciliome) can be divided into core and subtype-specific categories. Core genes whose reporters are analyzed in panel C and *Figure 1—figure supplement 2* are indicated by their function. (**B**) Lateral view of *C. elegans* hermaphrodite ciliated system. Sixty ciliated neurons from 25 different classes are distributed in five distinct anatomical regions. (**C**) Ciliome core components show persistent expression in double *daf-12(sa204); daf-19(m86)* mutants. The same extrachromosomal line was analyzed in the different genetic backgrounds. Each dot represents the total number of reporter-positive neurons in a single animal. Mean and standard error are represented. The mean number of remaining reporter-positive neurons in double *daf-12; daf-*

*Figure 1 continued on next page*

*Figure 1 continued*

19 mutants is indicated. Sample sizes for each genetic background: *che-11: n=8; osm-1: n≥7; ift-20: n≥9; che-13: n≥7; tub-1: n≥10; xbx-1: n≥5; peli-1: n≥8*. See *Figure 1—source data 1* for raw data and *Figure 1—figure supplements 1 and 2* for details on construct lengths and additional reporter scorings. (**D**) sc-RNA-seq data analysis identifies 10 TFs specifically enriched in ciliated sensory neurons. These TFs belong to FKH, ZF, NHR and HD families. See *Figure 1—figure supplement 3* for detailed description of TF expression in each ciliated neuron type. (**E**) ChIP-seq data analysis of 259 available TFs shows that FKH-8 ranks first in direct binding to regulatory regions assigned to the ciliome gene list. See *Figure 1—source data 2* for gene lists and *Figure 1—figure supplement 3* for core ciliome or subtype specific binding analysis. (**F**) Correlation of total number of peaks versus ciliome-gene peaks shows FKH-8 behaves as an outlier, demonstrating high binding to ciliome genes is not merely due to the high number of FKH-8 binding-events.

The online version of this article includes the following source data and figure supplement(s) for figure 1:

**Source data 1.** Raw quantification data of ciliome gene reporters in *daf-19(m86); daf-12(sa204)* included in *Figure 1* and *Figure 1—figure supplement 2*.

**Source data 2.** Gene lists and raw data for bioinformatics analysis in *Figure 1* and *Figure 1—figure supplement 3*.

**Figure supplement 1.** Ciliome reporters used in this work.

**Figure supplement 2.** Reporter expression for some core ciliome genes is abolished in *daf-19(m86)* mutant.

**Figure supplement 3.** Available -omics data identifies FKH-8 as a candidate transcriptional regulator of ciliome genes in *C.elegans*.

known to directly regulate ciliome gene expression in vertebrate motile-cilia cell types, rescue *fkh-8* mutant expression defects in *C. elegans*. This functional conservation is not observed with members of other FKH sub-families. Our results identify FKH-8 as the first TF acting together with RFX TFs in the direct regulation of the ciliome genes in sensory-ciliated cells and suggest that this function could be evolutionary conserved in vertebrates. Taken together, a global ciliome regulatory logic starts to emerge in which RFX and FKH TFs could act together in the direct regulation of ciliome gene expression both in cell types containing motile or primary cilia. Considering that ancestral eukaryotic cilium is proposed to combine motile and sensory functions, we speculate that RFX / FKH regulatory module might represent the ancestral state of eukaryotic ciliome gene regulation.

## Results

### Persistent activity of regulatory regions for ciliome genes in *daf-19/RFX* mutants

The activity of regulatory regions controlling ciliome gene expression is dramatically reduced in *daf-19(m86)* null mutants. However, for several ciliome reporters, some residual activity has been anecdotally reported (*Burghoorn et al., 2012*; *Chu et al., 2012*; *De Stasio et al., 2018*; *Efimenko et al., 2005*; *Haycraft et al., 2001*; *Swoboda et al., 2000*). As *daf-19* is the only RFX TF encoded in the *C. elegans* genome, we reasoned that persistent ciliome enhancer activity in *daf-19(m86)* null mutants would underscore the presence of additional TF families acting in concert with DAF-19. Based on previous data, we selected enhancers and built fluorescent reporters for ten phylogenetically conserved and broadly expressed core cilia components: five intraflagelar transport (IFT) genes (*che-11, osm-1, ift-20, che-13, osm-5*); the transition zone transmembrane genes *tmem-107* and *mks-1*; a Tubby family of bipartite transcription factors involved in receptor trafficking (*tub-1*); the dynein-component *xbx-1* and the ubiquitin protein ligase *peli-1* (*Figure 1A*). Human orthologs for several of these genes are linked to ciliopathies (*Horani and Ferkol, 2021*; *Mukhopadhyay et al., 2005*; *Thevenon et al., 2016*). All fluorescent reporters contain at least one experimentally validated X-box, the binding site for DAF-19/RFX (*Figure 1—figure supplement 1*). To avoid the dauer constitutive phenotype of *daf-19(m86)* null animals, we analyzed reporter expression in *daf-19(m86); daf-12(sa204)* double mutants and *daf-12(sa204)* was used as control, as previously reported (*De Stasio et al., 2018*; *Phirke et al., 2011*).

In *daf-12(sa204)* worms, all reporters show broad activity in the ciliated system, with mean reporter expression in at least 30 ciliated neurons, except for *mks-1* and *osm-5* reporters that showed expression in less than 20 cells, suggesting other enhancers outside the analyzed sequences might drive expression in additional ciliated neurons (*Figure 1C*, *Figure 1—figure supplement 2* and *Figure 1—source data 1*). As expected, *daf-19(m86); daf-12(sa204)* double mutants show a dramatic decrease in the number of neurons positive for each reporter (*Figure 1C* and *Figure 1—figure supplement*

*2*). Importantly, all reporters except *tmem-107*, *mks-1,* and *osm-5*, which correspond to the shortest constructs, show persistent expression in some neurons (*Figure 1C*, *Figure 1—figure supplement 2* and *Figure 1—source data 1*). We hypothesized that these short constructs might lack binding sites for additional TFs acting with DAF-19. Indeed, we find that shorter versions of *xbx-1* and *peli-1* reporter constructs are more affected by *daf-19* mutation than corresponding longer constructs, consistent with shorter sequences lacking additional regulatory information (*Figure 1C*). Altogether our data strongly suggests that additional TF or TFs act together with DAF-19 to directly activate core ciliome gene expression.

## Identification of FKH-8 as candidate regulator of ciliome gene expression

We reasoned that similar to *daf-19,* additional regulators of cilia gene expression could act broadly on many genes coding for ciliome components and in many different ciliated neuron types. Thus, to identify these putative candidates, we combined three strategies: *cis*-regulatory analysis of the ciliome genes, TF expression enrichment in the sensory ciliated system and TF binding to putative regulatory regions of the ciliome genes.

We built a manually curated list of 163 cilium effector genes (Materials and methods and *Figure 1—figure supplement 2*). This list can be divided in four categories: (1) 73 '*core components*' present in all types of cilia and thus expressed by all ciliated neurons in *C. elegans*. Core components include IFT particles, kinesins, dyneins, BBSome complex, etc; (2) 68 '*Subtype specific*' genes, that code for channels or receptors located in cilia that are expressed in a neuron-type-specific manner, providing neuron-type-specific functions; (3) 13 '*Broad expression*' genes, specifically expressed within the ciliated system but not associated with well-defined core cilia functions and (4) 9 '*Male*' genes that code for genes with male-specific cilia functions (*Figure 1—source data 2*).

De novo motif enrichment analysis using the promoters of ciliome genes identified previously known RFX consensus binding sites (X-box motif). In agreement with published results, X-box motifs are preferentially associated to '*Core*' and '*Broadly expressed*' ciliome genes (*Figure 1—figure supplement 3*; *Burghoorn et al., 2012*; *Efimenko et al., 2005*; *Swoboda et al., 2000*). An additional motif matching the pro-neural bHLH TF *lin-32*/Atoh1 is present in 28% of the genes, with no particular bias to any ciliome category (*Figure 1—figure supplement 3*). The pro-neural binding motif might reflect the neuronal nature of this gene set, as in *C. elegans* cilia are only present in neurons. Motif enrichment analysis limited only to the 102 genes containing predicted X-box sites identified two additional motifs one showing similarity to DAF-19/RFX binding site and the other to the previously reported C-BOX motif for whom TF binding has not been determined (*Burghoorn et al., 2012*; *Figure 1—figure supplement 3*). Analysis limited to the 61 genes lacking predicted X-box sites failed to identify enriched motifs for known transcription factors. Thus, motif enrichment of ciliome genes regulatory regions failed to pinpoint additional TF candidates to regulate ciliome gene expression. The X-box motif consists of a long imperfect palindromic consensus binding site with high information content (*Figure 1—figure supplement 3*), in contrast, most TF binding motifs (TFBM) are often short and degenerate, thus predicted matches can be found at high frequency in the genome. This feature might underlie the failure to find enriched motifs for additional TFs in ciliome gene regulatory regions.

As an alternative to motif enrichment analysis, we turned to TF expression enrichment. We hypothesized that TFs acting broadly on sensory cilia gene expression could show enriched expression in the sensory ciliated neurons. Using available single cell RNA expression data (sc-RNAseq) from the second larval stage (*Cao et al., 2017*), we retrieved the expression pattern of 861 *C. elegans* transcription factors (*Narasimhan et al., 2015*). Ten transcription factors are specifically enriched within the ciliated sensory neurons compared to other neuron types or non-neuronal tissues (*Figure 1D*). Not surprisingly, *daf-19* expression is not enriched in ciliated neurons, as *daf-19* is expressed panneuronally and only a specific splicing isoform is restricted to ciliated neurons (*Senti et al., 2008*). The expression of TFs controlling terminal identity is often maintained throughout the life of the animal. Using an independent set of sc-RNAseq data from young adult (*Taylor et al., 2021*), we found that among the 10 TF candidates only FKH-8 expression is detected in all 25 different types of sensory ciliated neurons (*Figure 1—figure supplement 3*), suggesting it could be a good candidate to act together with DAF-19 in the regulation of ciliome gene expression.

Finally, we interrogated 446 published ChIP-seq datasets (*Luo et al., 2020*), corresponding to 259 different TFs (including FKH-8 but not DAF-19), for nearby binding to the ciliome gene list (*Figure 1—source data 2*). We find FKH-8 behaves very differently from the rest of TFs with at least one FKH-8 binding peak associated to 49% of the genes on the ciliome gene list (*Figure 1E*). FKH-8 binds both core components and subtype-specific ciliome genes (*Figure 1—figure supplement 3*), although similar to X-box motifs, FKH-8 binding is significantly more prevalent for core ciliome genes (75% compared to 22% binding to subtype genes). Thus, both sc-RNAseq and ChIPseq data analysis pinpoint FKH-8 as a good candidate TF to directly control ciliome gene expression.

## FKH-8 is expressed in all ciliated sensory neurons

FKH-8::GFP fosmid expression at young adult stage is detected in all ciliated sensory neurons, as assessed by co-localization with the *ift-20* core ciliome reporter (*Figure 2A and B* and *Figure 2—source data 1*). Three non-ciliated neurons, PVD, VC4 and VC5 show FKH-8 expression, while no expression is detected in non-neuronal tissues. *C. elegans* male nervous system contains additional ciliated sensory neurons, mostly in the tail, which also express FKH-8 (*Figure 2B*). During embryonic development, there is a similar overlap between FKH-8 and *ift-20* reporters (*Figure 2—figure supplement 1*). Correlation between *fkh-8*, *ift-20* and *daf-19* expression during development is also observed using Uniform Manifold Approximation and Projection (UMAP) representation of embryonic sc-RNA-seq data (*Packer et al., 2019*; *Figure 2—figure supplement 1*). In addition, there is a high gene expression correlation for the 73 core ciliome genes and *daf-19* or *fkh-8* expression but not with other TFs (*Figure 2—figure supplement 1*). Thus, our analysis shows that FKH-8 is expressed almost exclusively in the whole ciliated sensory system and its developmental expression correlates with core ciliome gene expression.

## FKH-8 binds near X-boxes associated to ciliome genes

Next, we extended FKH-8 ChIPseq data analysis to the whole genome. FKH-8 binds a total of 5035 genomic regions assigned to 3987 genes. Most peaks are associated to promoter regions (58.65%). Gene ontology analysis of FKH-8 bound genes shows enrichment for cilia functions or dauer regulation (which is also dependent on cilia integrity; *Figure 2C*, *Figure 2—source data 2*).

DNA consensus motifs bound by FKH-8 have not been experimentally determined. FKH TF family binds the canonical consensus RYMAAYA (*Pierrou et al., 1994*) and an alternative motif, termed FKH-like (FHL), characterized by a GACGC core sequence (*Nakagawa et al., 2013*). De novo motif enrichment analysis of FKH-8 ChIP-seq peaks does not show any match for FKH canonical binding site but identifies a motif that highly resembles the FHL motif (*Figure 2D*). This motif, present in 27% of the peaks, is enriched at central positions suggesting it could act as FKH-8 primary binding motif (*Figure 2D*).

We noticed that eight out of the 12 functional X-boxes present in the core ciliome gene reporters analyzed in *Figure 1C* overlap with FKH-8 ChIP-seq peaks (*Figure 1—figure supplement 1*). Thus, we next looked for X-box enrichment in FKH-8 bound regions. 21% of FKH-8 peaks contain at least one match for the DAF-19 position weight matrix (*Figure 2E*). Importantly, X-boxes are preferentially found also at central locations, suggesting they could be in close proximity to FKH-8-bound sites (*Figure 2E*). X-boxes are less significantly or not significantly enriched in ChIP-seq datasets for other FKH TFs (*Figure 2—figure supplement 2* and *Figure 2—source data 2*), which is consistent with specific co-binding of DAF-19 and FKH-8.

Next, we analyzed the presence of X-boxes and FKH-8 binding events specifically associated to ciliome genes. Regulatory sequences for 34% of ciliome genes contain both X-box motifs and FKH-8 genomic binding (*Figure 2F* and *Figure 2—source data 2*). This dual regulatory signature is more prevalent in core ciliome genes (present in 62% of this gene category) than in subtype-specific ciliome (present in 6% of these genes) (*Figure 2F* and *Figure 2—source data 2*). In addition, most X-boxes are located less than 600 bp from the center of its closest FKH-8 ChIP-seq peak (*Figure 2G*), this close proximity is a feature of core-ciliome components (found in 66% of core ciliome genes) but not for sub-type ciliome genes (only 9% of X-boxes found in this category contain a nearby FKH-8 peak) (*Figure 2—figure supplement 2* and *Figure 2—source data 2*).

Finally, we assessed FKH-8 and DAF-19 physical interaction by co-expression in HEK293 cells and co-immunoprecipitation. Our results show that FKH-8 and DAF-19 interact bound to DNA

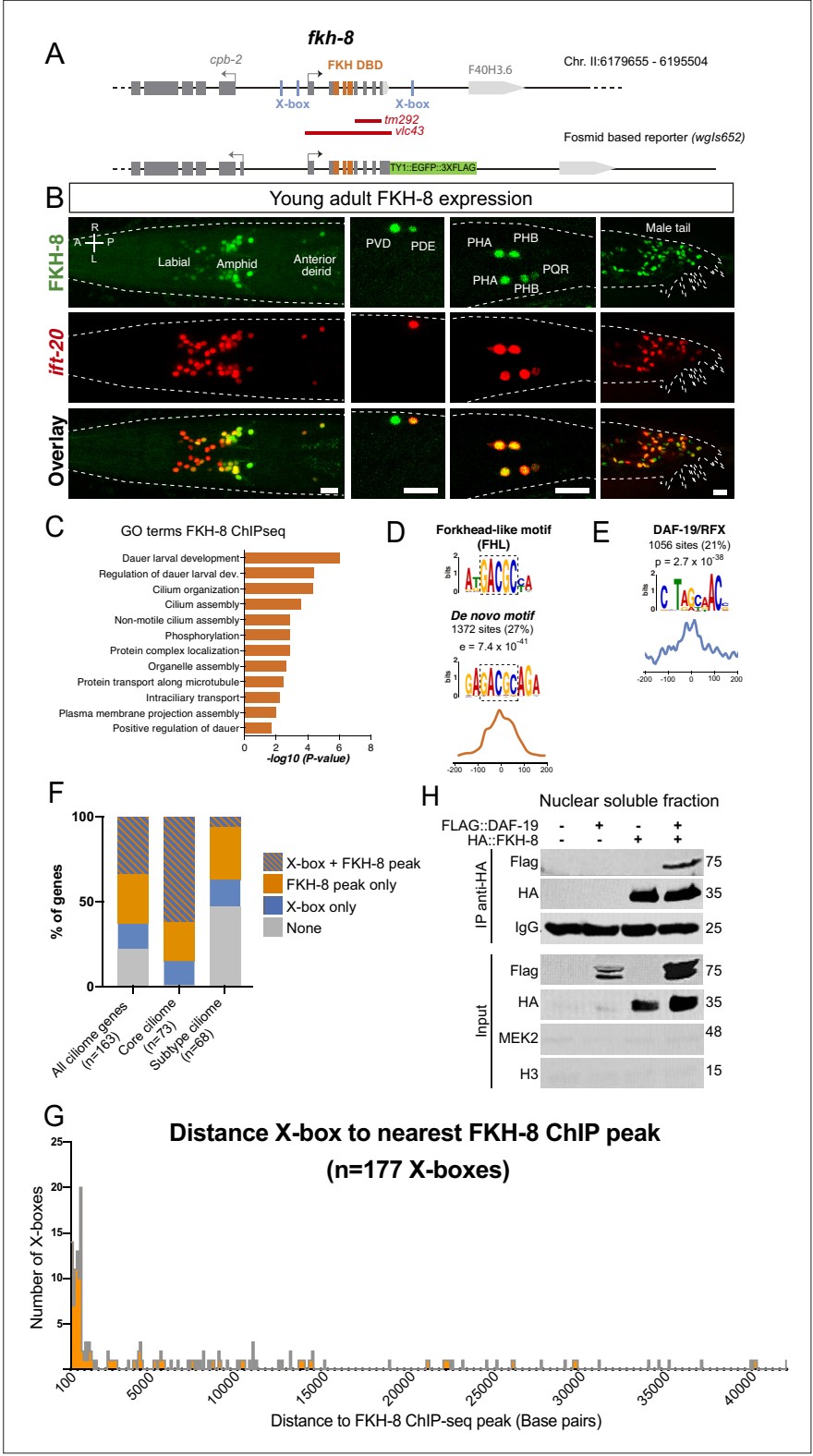

**Figure 2.** FKH-8 is expressed in sensory ciliated neurons, binds ciliome genes near DAF-19 X-boxes and physically interacts with DAF-19. (**A**) *fkh-8* locus (top) and fosmid based *fkh-8* reporter (bottom). Grey boxes represent exons and orange boxes correspond to exons coding for the FKH DNA binding domain (DBD). Putative *daf-19*/RFX binding sites (X-boxes) are depicted with blue lines. Red bars indicate extension for the corresponding deletion alleles. (**B**) Dorso-ventral views of young adult animals expressing both the fosmid-based FKH-8::GFP reporter

*Figure 2 continued on next page*

*Figure 2 continued*

(in green) and an integrated reporter for the panciliary marker *ift-20* (in red). A: anterior, P: posterior, R: right, L: left. Scale bar = 10 μm. See *Figure 2—source data 1* for quantification and *Figure 2—figure supplement 1* for embryonic expression patterns and expression correlation with DAF-19 and ciliome genes. (**C**) Genes associated to nearby FKH-8 binding events enrich Gene Ontology terms related to cilia regulated processes and/or functions. Data correspond to adjusted p-value. See *Figure 2—source data 2* for gene lists associated to GO terms (**D**) De novo motif analysis of FKH-8 ChIP-seq data identifies a motif present in 27% of peaks, enriched at central positions, that matches a Forkhead like (FHL) motif. (**E**) DAF-19/RFX binding motifs (PWM M1534_1.02) are present in 21% of the FKH-8 bound regions and are enriched at central positions. See *Figure 2—figure supplement 2* for similar analysis on additional FKH ChIP-seq data sets. (**F**) Distribution of ciliome genes in four different categories: (1) genes with both X-box motifs and FKH-8 binding events; (2) genes with only FKH-8 binding; (3) Genes with X-box motifs only and (4) Genes with neither FKH-8 binding or X-boxes. Most ciliome genes contain both X-boxes and FKH-8 peaks, this dual signature is highly prevalent in core ciliome genes while is minoritary in subtype ciliome genes. See *Figure 2—source data 2* for gene lists associated to each signature. (**G**) Distance between X-boxes found in ciliome genes and the center of the nearest FKH-8 ChIP-seq peak. 42% of X-boxes are located less than 600 bp from a FKH-8 ChIP-seq peak. See *Figure 2—figure supplement 2* for differential analysis of core and subtype ciliome genes. (**H**) Co-immuno precipitation of HA tag FKH-8 and FLAG tag DAF-19 expressed in HEK293 cells shows physical interaction between both transcription factors in the soluble fraction of nuclear extracts. MEK2 is used to assess for the presence of cytoplasmic components and Histone H3 to assess the presence of chromatin. See *Figure 2—source data 3* for original blots and *Figure 2—figure supplement 2* for additional analysis of interaction in chromatin associated fractions.

The online version of this article includes the following source data and figure supplement(s) for figure 2:

**Source data 1.** Raw quantification data of FKH-8 and ift-20 reporter co-expression represented in *Figure 2B*.

**Source data 2.** Raw data for bioinformatics analysis in *Figure 2* and *Figure 2—figure supplements 1 and 2*.

**Source data 3.** Co-IP original files of unedited gels in *Figure 2*.

**Figure supplement 1.** *fkh-8* expression along ciliated system development.

**Figure supplement 2.** FKH-8 binds near X-BOX motifs.

**Figure supplement 2—source data 1.** Co-IP original files of unedited gels for the chromatin fraction.

---

(*Figure 2—figure supplement 2* and ) and more importantly also in the nuclear soluble fraction free of chromatin (*Figure 2H* and *Figure 2—source data 3*). In summary, our data is consistent with FKH-8 and DAF-19 acting together to regulate ciliome gene expression, particularly in core cilia components.

## *fkh-8* mutants show defects in ciliome reporter gene expression

The only available *fkh-8* mutation, *tm292*, is a deletion downstream the FKH DNA binding domain, suggesting it might not be a null allele (*Figure 2A*, *Figure 2—source data 1*). Thus, we built *fkh-8(vlc43)*, a deletion allele that removes the whole *fkh-8* coding region (*Figure 2A*). We selected eight reporters for six genes that code for core cilia components and that overlap with FKH-8 ChIP-seq peaks (*Figure 1—figure supplement 1*) and analyzed their expression in *fkh-8(tm292)* and *fkh-8(vlc43)* mutants.

Both *fkh-8* mutant alleles show significant expression defects in all reporters except for *tub-1*/Tub and the long *peli-1*/Peli1,2,3 and *xbx-1* reporters (*Figure 3A and B Figure 3—figure supplement 1* and *Figure 3—source data 1*). Lack of fluorescence reporter expression in *fkh-8* mutants reflects enhancer activity defects and not the absence of the ciliated neurons per se, as *tub-1*/Tub and the long *peli-1*/Peli1,2,3 reporters are expressed in 53 and 46 ciliated neurons respectively in *fkh-8* mutants, similar to *wild type* expression levels (*Figure 3—figure supplement 1* and *Figure 3—source data 1*). Phenotypes are often more penetrant in *fkh-8(vlc43)* null allele than in the *fkh-8(tm292)* and both *fkh-8(vlc43)* and *fkh-8(tm292)* heterozygote animals show similar reporter expression levels as *wild type* indicating both alleles are recessive and *tm292* is a hypomorph (*Figure 3—figure supplement 1*).

Endogenously tagged *osm-5* core cilome gene [*osm-5(syb6528), osm-5::SL2::GFP::H2B*] shows panciliary expression in *wild type* animals (*Figure 3C*), fluorescence intensity is greatly reduced in *fkh-8(vlc43)* animals (*Figure 3C and D*) further supporting the role of FKH-8 in direct control of ciliome gene expression.

*fkh-8(vlc43)* animals show missing *ift-20* expression in ten neurons including the four pairs of dopaminergic ciliated mechanosensory neurons (CEPV, CEPD, ADE and PDE). Expression in *fkh-8(vlc43)*

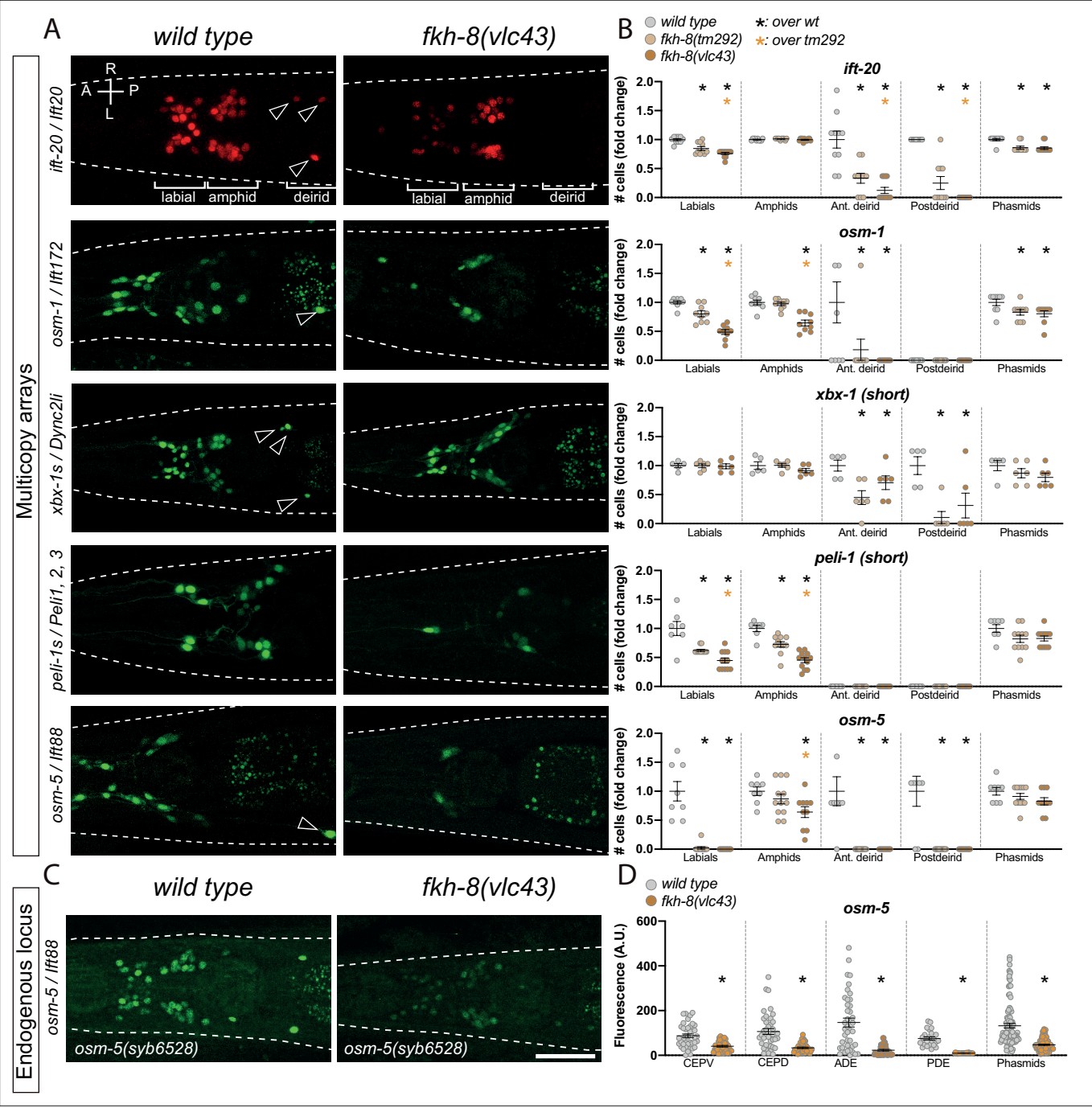

**Figure 3.** FKH-8 TF and FKH-binding sites are required for correct core ciliome gene reporter expression. (**A**) Dorso-ventral images from young adult heads expressing different core ciliome multicopy array gene reporters in *wild type* and *fkh-8(vlc43)* null mutant animals. All reporters are extrachromosomal arrays except for *ift-20* reporter which is integrated. Arrow heads point deirid expression lost in the mutant. A: anterior, P: posterior, R: right, L: left. (**B**) Quantification of the number of *gfp*-positive cells in five distinct anatomical regions for each reporter in *wild type, fkh-8(tm292)* hypomorphic allele and *fkh-8(vlc43)* null mutant. To facilitate comparisons, values in each region are normalized to controls. The same extrachromosomal line was analyzed in the different genetic backgrounds. Each dot represents the number of reporter-expressing neurons scored in a single animal. Mean and standard error are represented. Black asterisk denotes significantly different from *wild type* and orange asterisk indicates *vlc43* is significantly different from *tm292* allele. Sample sizes for each genetic background: *ift-20*: n≥10; *osm-9*: n=9; *osm-5*: n≥8; *peli-1*: n≥7; *xbx-1*: n≥5. See *Figure 3—source data 1* for raw scoring data, *Figure 3—figure supplement 1* for analysis of the hypomorphic recessive nature of the *tm292* allele and quantification of additional reporters not affected in *fkh-8* mutants, see *Figure 3—figure supplement 2* and *Figure 3—source data 2* for functional characterization of predicted FKH binding sites in *ift-20* and *xbx-1* regulatory regions. (**C**) Dorso-ventral images from young adult heads expressing *GFP* from the endogenously tagged *osm-5* locus [*osm-5(syb6528), osm-5::SL2::GFP::H2B*] in *wild type* and *fkh-8(vlc43)* null mutant. A global decrease

*Figure 3 continued on next page*

*Figure 3 continued*

in fluorescence intensity is detected in *fkh-8(vlc43)* animals compared to *wild type*. Scale bar = 25 μm. (**D**) Fluorescence intensity level quantification in specific ciliated neuron populations shows significant reduction of expression in *fkh-8(vlc43)* animals. A. U.: arbitrary units. See *Figure 3—source data 1* for raw scoring data. n≥*20* for each cell type and genetic background.

The online version of this article includes the following source data and figure supplement(s) for figure 3:

**Source data 1.** Raw quantification data of ciliome gene reporter expression defects in *fkh-8* mutants and cis-regulatory point mutation analysis corresponding to *Figure 3* and *Figure 3—figure supplements 1 and 2*.

**Source data 2.** Motif enrichment analysis of *xbx-1* and *ift-20* reporters.

**Figure supplement 1.** *f. fkh-8(tm292)* is a hypomorphic recessive allele.

**Figure supplement 2.** *f.* Functional characterization of putative FKH sites in cis-regulatory modules of two core ciliome components.

---

animals of *fkh-8* cDNA under the control of a *dat-1* dopaminergic specific promoter, which is unaffected in *fkh-8* mutants (see below Figure 7 and *Figure 7—source data 1*), is able to rescue *ift-20* reporter expression, consistent with a cell autonomously role of *fkh-8* in the regulation of ciliome gene expression.

Next, we complemented the TF mutant analysis with *cis*-regulatory mutant analysis. We focused on *ift-20* and short *xbx-1* reporters which both overlap with FKH-8 ChIP-seq peaks (*Figure 3—figure supplement 2*). Three independent transgenic lines with point mutations for FKH binding sites show broad expression defects both for *ift-20* and *xbx-1* reporters (*Figure 3—figure supplement 2* and *Figure 3—source data 1 and 2*). *cis*-mutation expression defects are stronger than the ones observed for *fkh-8* mutant alleles suggesting either other FKH factors can compensate the lack of *fkh-8* or that *cis*-mutations could affect the binding of other TFs in addition to FKH-8. Future work will be required to assess if other FKH TFs are expressed in specific subpopulations of ciliated neurons and if they can compensate for the lack of FKH-8.

In summary, our *cis* regulatory and *fkh-8* mutant analyses together with ChIP-seq data unravel a cell autonomous role for FKH-8 in the direct regulation of ciliome gene expression.

## FKH-8 and DAF-19/RFX act synergistically

FKH-8 binds five different locations in the *daf-19* locus (*Figure 4A*) while *fkh-8* locus contains 3 putative X-box sites (*Figure 2A*), suggesting they could cross-regulate each other's expression. Transcription of *daf-19* generates different isoforms that share the carboxyl terminal (Ct) domain and the RFX DNA binding domain but differ in the amino-terminal region (*Figure 4A*). Some of these isoforms are expressed in a mutually exclusive manner: *daf-19d* is specifically expressed in ciliated neurons while *daf-19a/b* isoforms are expressed in the rest of the nervous system but not in ciliated neurons (*Senti et al., 2008*). Accordingly, a fosmid-based Ct-tagged DAF-19 reporter that labels all isoforms is broadly expressed in neurons (*Figure 4—figure supplement 1*). We did not find any obvious DAF-19::GFP expression defects in *fkh-8(vlc43)* mutants (*Figure 4—figure supplement 1*). Co-localization of DAF-19::GFP with *dat-1::mcherry* dopaminergic reporter expression or DiD lipophilic staining also reveals similar expression in *wild type* and *fkh-8(vlc43)* mutants in the dopaminergic or amphid ciliated neurons (*Figure 4—figure supplement 1*). Thus, our data suggest that, despite its extensive binding to *daf-19* locus, FKH-8 does not seem to be required for *daf-19* expression in ciliated neurons.

Next, we assessed FKH-8::GFP fosmid expression in *daf-19(m86); daf-12(sa204)* double mutant. In contrast to ciliated-neuron-specific expression pattern seen in *wild type*, FKH-8::GFP is expressed seemingly pan-neuronally in *daf-19(m86); daf-12(sa204)* double mutants (*Figure 4B*), suggesting a repressive role for DAF-19. FKH-8::GFP expression in the PDE dopaminergic ciliated sensory neuron is unaffected in *daf-19(m86); daf-12(sa204)* double mutants as assessed by co-localization with *dat-1:cherry* [91% PDE neurons are FKH-8::GFP positive in *wild type* animals and 92% in *daf-19(m86); daf-12(sa204)* mutants], suggesting that FKH-8::GFP expression is unaffected by the lack of *daf-19* in ciliated neurons (*Figure 4—source data 1*). *daf-19(m86)* allele affects all isoforms; as DAF-19 isoform D is expressed in ciliated neurons, our results suggest DAF-19D is not necessary for FKH-8 expression in ciliated neurons. In contrast, DAF-19 isoforms A and B seem to repress FKH-8 expression in non-ciliated neurons, in agreement with this hypothesis, *daf-19(of5),* a mutant allele that specifically affects *daf-19 a/b* isoform expression (*Figure 4A*) shows similar pan-neuronal de-repression of FKH-8::GFP (*Figure 4B*). Consistent with the role for DAF-19A/B as repressor (*De Stasio et al., 2018; Senti*

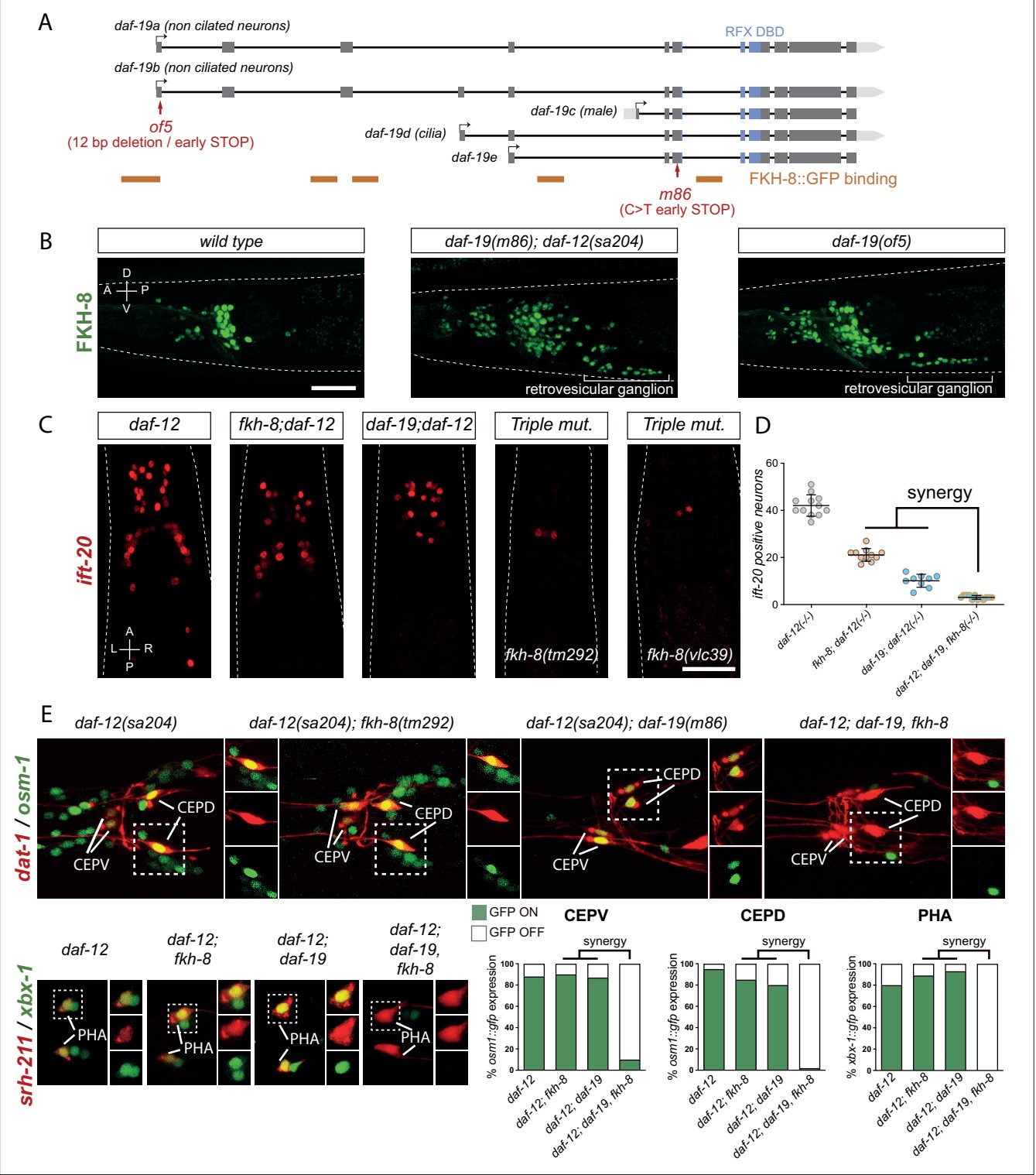

**Figure 4.** FKH-8 and DAF-19 exhibit crosstalk and synergistic effects in the transcriptional regulation of ciliome genes. (**A**) *daf-19* locus codes for five different *daf-19* isoforms. Grey boxes represent exons whereas blue boxes correspond to exons coding for the RFX DNA binding domain (DBD). FKH-8 binding events are depicted as orange lines. Red arrows locate mutations of the corresponding alleles. (**B**) Lateral views from young adult hermaphrodite heads expressing *fkh-8* fosmid-based reporter (*wgIs652*). Lack of all *daf-19* isoforms (*m86* allele) derepresses *fkh-8* in non-ciliated neurons. This phenotype is mimicked by the specific absence of long *daf-19a/b* isoforms (*of5* allele). Scale bar = 25 μm. See ***Figure 4—figure supplement 1*** for unaffected DAF-19 expression in ciliated neurons in *fkh-8(vlc43)* mutants. (**C**) Dorso-ventral images from young adult hermaphrodites showing core ciliome *ift-20* reporter expression in different genetic backgrounds. Scale bar = 25 μm. (**D**) Mean number of *ift-20* reporter-expressing

*Figure 4 continued on next page*

*Figure 4 continued*

neurons in *daf-12(sa204); daf-19(m86), fkh-8(tm292)* triple mutants is significantly different from each of the double mutants and significantly lower than the expected from the multiplicative effect of both *daf-12(sa204); fkh-8(tm292)* and *daf-12(sa204); daf-19(m86)* animals. The same extrachromosomal line was analyzed in the different genetic backgrounds. Each dot represents the number of reporter-expressing neurons scored in a single animal. Mean and standard error are represented. See *Figure 4—figure supplement 2* for quantification of FKH-8 and DAF-19 synergistic effects in *xbx-1* and *peli-1* reporter expression and *Figure 4—source data 1* for raw data and statistics for all analyzed genetic backgrounds. n≥10 for each genetic background. (**E**) Analysis of *osm-1* and *xbx-1* ciliome reporters in specific subpopulations of ciliated neurons. CEPV and CEPD are labeled with *dat-1::mcherry (otIs181)* and *srh-211:: tagRFP (vlcEx1365)* is expressed in PHA neuron, both reporters are unaffected in all genetic backgrounds. Quantification of ciliome reporters is depicted in the corresponding graphs. *Figure 4—source data 1* for raw data and statistics for all analyzed genetic backgrounds, n=30 worms per genotype and reporter construct.

The online version of this article includes the following source data and figure supplement(s) for figure 4:

**Source data 1.** Raw quantification data of synergistic actions of fkh-8 and daf-19 depicted in *Figure 4*, *Figure 4—figure supplements 1 and 2*.

**Figure supplement 1.** Lack of FKH-8 has no major effect on DAF-19 expression.

**Figure supplement 2.** FKH-8 and DAF-19 show synergistic effects in the transcriptional regulation of the ciliome.

*et al., 2008*), we also found *kap-1* ciliome gene reporter is de-repressed in *daf-19(m86); daf-12(sa204)* mutants (*Figure 4—figure supplement 2*). Interestingly, both *kap-1* expression in ciliated neurons as well as *kap-1* de-repression in *daf-19* mutants is independent of FKH-8 (*Figure 4—figure supplement 2*). In AWB neurons, *kap-1* reporter expression is regulated by FKH-2 transcription factor which is a downstream target of DAF-19 (*Mukhopadhyay et al., 2007*). We find *fkh-2* fosmid reporter (*wgIs185*) expression in AWB is also independent of FKH-8 (*Figure 4—source data 1*) further supporting that *kap-1* expression in *wild type* and de-repression in *daf-19* mutants is independent of FKH-8.

Altogether our data supports previous results on the repressive role of DAF-19 A/B long isoforms (*De Stasio et al., 2018*; *Senti et al., 2008*), which at least in part seems to be independent of FKH-8 activity.

Next, we assessed sufficiency of DAF-19 and FKH-8 for the expression of *ift-20* core ciliome gene reporter. Using a heat shock inducible promoter, we ectopically expressed DAF-19D during embryonic development. DAF-19D induction was sufficient to produce ectopic *ift-20* expression while similar FKH-8 expression did not produce any significant effect (*Figure 4—figure supplement 2*). These results, together with the stronger phenotypes of *daf-19(m86)* mutants, indicate that although both TFs are required for ciliome gene expression, RFX TFs display more instructive functions in ciliome gene expression.

Considering that DAF-19 and FKH-8 share several ciliome gene targets, bind the regulatory regions of ciliome genes in close proximity and both can physically interact, we aimed to assess if DAF-19 and FKH-8 cooperate in the regulation of ciliome gene expression. *daf-19* and *fkh-8* genes are both located in chromosome II, despite several attempts, we failed to generate *daf-19(m86), fkh-8(vlc43) II; daf-12(sa204)* triple mutants but we were able to obtain *daf-19(m86), fkh-8(tm292); daf-12(sa204)* recombinant animals. To study synergistic effects, we counted the number of reporter expressing cells in the different genetic backgrounds. Loss of *ift-20, peli-1,* and *xbx-1* reporter expression in triple mutants is greater than what will be expected for the multiplicative effect of the individual *daf-19(m86); daf-12(sa204)* and *fkh-8(tm292); daf-12(sa204)* phenotypes (*Figure 4C and D* and *Figure 4—figure supplement 2*) suggesting FKH-8 and DAF-19 could act synergistically. Of note, these reporters still show some vestigial expression in the triple mutant (*Figure 4C* and *Figure 4—figure supplement 2*). We CRISPR-engineered a full deletion of the *fkh-8* locus in the *daf-19(m86); daf-12(sa204); ift-20::rfp* strain which generated a viable triple null mutant [*fkh-8 (vlc39)* allele]. These animals show similar residual *ift-20* expression in a couple neurons (*Figure 4C*), based on location we tentatively identified them as CEPV or BAG neurons, however lack of co-expression using specific reporters (*otIs259 [dat-1::gfp]* and *rpIs3 [gcy-33p::GFP]*) discard these neuron identities. Taking into consideration that *daf-19(m86)* mutants show ectopic expression for some ciliome genes and the absence of other ciliated neuron candidates in the region, we conclude *ift-20*-positive cells in triple mutants are not likely to be ciliated neurons.

Importantly, although our analysis of total number of cells suggest synergy between FKH-8 and DAF-19, ectopic reporter expression in *daf-19(m86)* mutants could be masking additional synergistic effects. Thus, to unequivocally determine synergy, we labeled specific ciliated neuron populations in the different genetic backgrounds. We find *osm-1* reporter expression in CEPV and CEPD neurons

is unaffected or only slightly affected in *fkh-8(tm292); daf-12(sa204) and daf-19(m86); daf-12(sa204)* double mutants compared to *daf-12(sa204)* controls but is completely abolished in triple mutants (*Figure 4E* and *Figure 4—source data 1*). Similarly, *xbx-1* expression in PHA neurons is only significantly affected in triple mutants (*Figure 4E* and *Figure 4—source data 1*). These data strongly suggest FKH-8 and DAF-19 act synergistically in the expression of specific ciliome genes and cellular contexts.

## FKH-8 is required for correct cilia morphology

Mutations in several ciliome core components, including *osm-5* and *xbx-1*, whose reporters are affected in *fkh-8* mutants, show cilium morphology defects (*Blacque et al., 2004*; *Mukhopadhyay et al., 2007*; *Perkins et al., 1986*; *Starich et al., 1995*). In addition, cilia length is controlled by a balance between cilia assembly and disassembly regulated by IFT and mutants for ciliome components can produce both shortened or elongated cilia (*Blacque et al., 2004*; *Burghoorn et al., 2007*; *Fujiwara et al., 1999*). One of the most commonly used methods to assess gross cilium integrity is lipophilic dye staining (like DiD), which in *wild type* animals labels a subpopulation of amphid and phasmid neurons (*Starich et al., 1995*). Despite ciliome gene expression defects, *fkh-8(vlc43)* animals show similar DiD staining compared to *wild type* (*Figure 4—source data 1*).

Next, we directly analyzed cilium morphology labeling specific subpopulations of ciliated neurons (*Figure 5*, *Figure 5—source data 1*). Cilium length in CEP and AWB neurons is significantly reduced in *fkh-8(vlc43)* mutants compared to controls, while ADF neuron cilium length is significantly increased (*Figure 5*, *Figure 5—source data 1*). In addition, *fkh-8* mutants display arborization defects in AWA cilia (*Figure 5*, *Figure 5—source data 1*). Of note, *osm-5* kinesin, which our data show is a direct target for FKH-8, is required for correct AWB cilia morphology (*Mukhopadhyay et al., 2007*), which is also affected in *fkh-8* mutants, suggesting both phenotypes could be correlated.

To discard cilia morphology defects were caused by selective effects in the mutants of sodium azide (used to paralyze worms) we treated worms with levamisole. Unexpectedly, we found *fkh-8(vlc43)* mutants are resistant to this drug. Of note, *daf-19* mutants are also levamisole resistant (*Senti et al., 2008*), which suggest this phenotype for both mutants could be related to their ciliogenic functions. Alternatively, we used polystyrene beads to physically immobilized worms. We found that cilia morphology defects were also present in *fkh-8(vlc43)* mutants using this method (*Figure 5—figure supplement 1*). Thus, FKH-8 is necessary to regulate correct cilium length and morphology in diverse types of ciliated neurons.

## *fkh-8* mutants display defects in a wide range of cilia mediated behaviors

In *C. elegans* cilia are necessary to mediate sensory functions (*Bargmann, 1993*); thus, we interrogated *fkh-8* mutants with a battery of sensory paradigms.

*fkh-8* mutants respond similarly to *wild type* animals to body gentle touch stimuli, which are mediated by not ciliated neurons (*Chalfie and Sulston, 1981*; *Figure 6—figure supplement 1*), discarding general response or locomotory defects in *fkh-8* mutants. Response to posterior harsh touch, which is redundantly mediated by ciliated PDE and non-ciliated PVD neurons (*Li et al., 2011*) is also unaffected in *fkh-8(tm292)* and *fkh-8(vlc43)* animals, suggesting FKH-8 is not required to mediate this mechanosensory behavior (*Figure 6—figure supplement 1*).

We tested two additional mechanosensory behaviors mediated only by ciliated sensory neurons: basal slowing response, mediated by dopaminergic ciliated neurons (*Sawin et al., 2000*) and nose touch, mediated by ASH, FLP and OLQ ciliated neurons (*Kaplan and Horvitz, 1993*). No defects on basal slowing response are found in *fkh-8(vlc43)* null mutants, while both *fkh-8* alleles are defective for nose touch responses (*Figure 6A and B*, *Figure 6—source data 1*). *vlc43* null allele shows stronger defects than *tm292* allele, supporting the hypomorphic nature of *tm292* allele (*Figure 6A*, *Figure 6—source data 1*). *fkh-8(vlc43)* animals are slightly but significantly dauer constitutive at 27 °C compared to N2 controls (*Figure 6C*, *Figure 6—source data 1*), which might indicate *fkh-8* mutants show defects in preventing dauer entry, a process mediated by ADF, ASI, and ASG ciliated neurons (*Bargmann and Horvitz, 1991*). Moreover, exposure to pheromones induces dauer entry in *fkh-8(vlc43)* animals less efficiently than in *wild type* animals [sixfold induction in *wild type* versus threefold induction in *fkh-8(vlc43)* animals; *Figure 6C*, *Figure 6—source data 1*], suggesting FKH-8 could also be required for correct dauer entry, which is mediated by ASJ ciliated neuron (*Bargmann and Horvitz, 1991*).

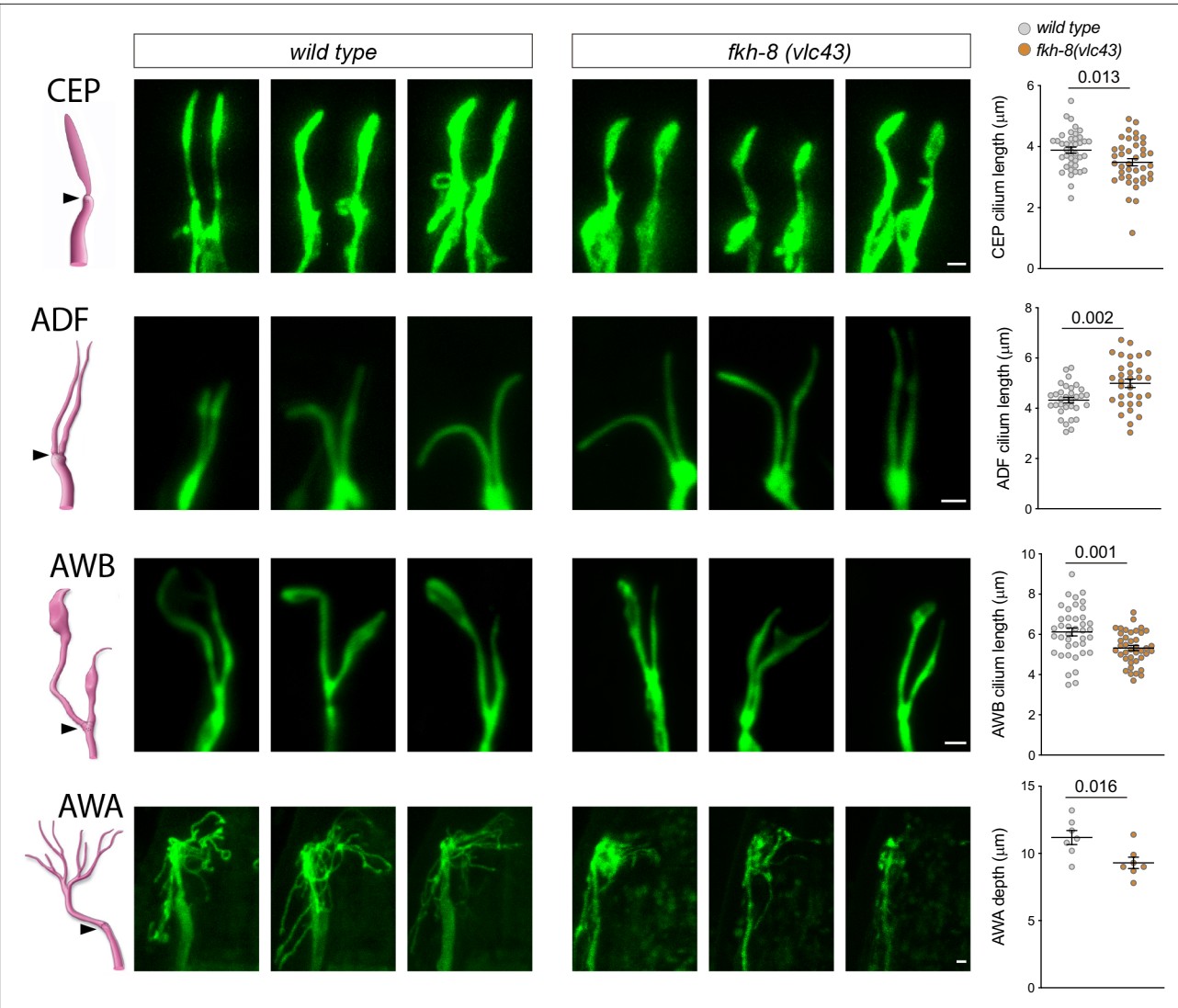

**Figure 5.** *fkh-8(vlc43)* null mutants display morphological defects in cilia. Integrated reporters unaffected in *fkh-8* mutant are used to label the cilia of several distinct subpopulations of ciliated neurons. CEP: *otIs259(dat-1::gfp)*; ADF: *zdIs13(tph-1::gfp)*; AWB: *kyIs104(str-1::gfp)*; AWA: *pkIs583(gpa-6::gfp)*. Panels show representative images from three animals in *wild type* and *fkh-8(vlc43)* mutant backgrounds. Cilium length of CEP and AWB neurons is significantly reduced in the absence of FKH-8 whereas ADF cilia length is increased. Depth of AWA cilium arborization is significantly reduced in *fkh-8(vlc43)* null mutants. Each dot in the graphs represents measures for a single cilium. Mean and standard error are represented. See *Figure 5—figure supplement 1* for cilia morphology analysis in worms immobilized in polystyrene beads and *Figure 5—source data 1* for raw data and statistics. CEP and AWB n=40; ADF n=32; AWA n=7.

The online version of this article includes the following source data and figure supplement(s) for figure 5:

**Source data 1.** Raw quantification data of cilia morphology analysis depicted in *Figure 5* and *Figure 5—figure supplement 1*.

**Figure supplement 1.** *fkh-8(vlc43)* null mutants display morphological defects in cilia using physical immobilization with polystyrene beads.

*fkh-8(vlc43)* null mutants, but not *tm292* allele, show significant odor avoidance defects to 2-nonanone (AWB mediated) and defective odor attraction to 2-heptanone (mediated by AWC; *Figure 6D and E*, *Figure 6—source data 1*; *Bargmann et al., 1993*; *Troemel et al., 1997*). Diacetyl attraction, which is mediated by AWA (*Sengupta et al., 1996*), is also decreased in *fkh-8(vlc43)* animals, although not significantly, due to high standard deviation values (*Figure 6F*, *Figure 6— source data 1*).

Finally, we tested gustatory responses to NaCl, Sodium Dodecyl Sulfate (SDS) and copper. *fkh-8* mutants are attracted to NaCl similar to N2 controls, a response that is mediated mainly by ASE ciliated neurons (*Bargmann and Horvitz, 1991*; *Figure 6G*, *Figure 6—source data 1*). In contrast, avoidance

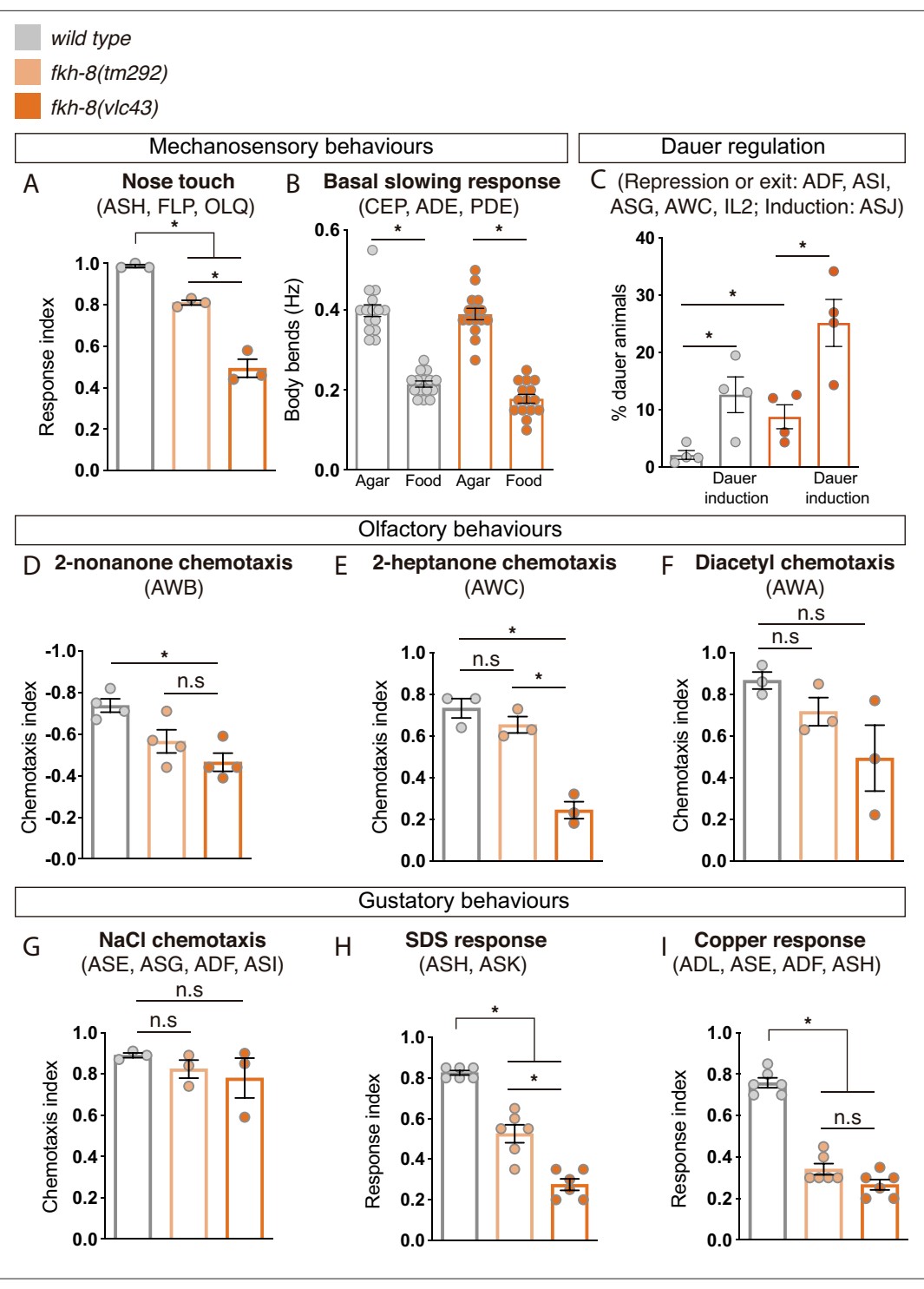

**Figure 6.** FKH-8 is required for the correct display of several sensory mediated behaviors. (**A**) Mutations in *fkh-8* significantly impair appropriate backward response to nose touch, revealing functionality defects for the ASH, FLP and/or OLQ ciliated neurons. This phenotype is stronger in *fkh-8(vlc43)* null mutants than in the hypomorphic *tm292* allele. n=20 animals per replicate, three biological replicates per genotype. (**B**) Decrease in locomotory rate upon re-entering a bacterial lawn is unaffected in *fkh-8* mutants. n=15 worms per genotype and condition. (**C**) *fkh-8* null mutants significantly fail to prevent *dauer* entry. Pheromones induce *dauer* in *fkh-8* mutants, albeit less efficiently than in controls. Four biological replicates n>295 per replicate and genotype. (D to F) Lack of *fkh-8* significantly impairs olfaction-mediated behaviors. Defects are observed for 2-nonanone repulsion mediated by AWB [*Wild type* n=59, 128, 114, 165; *fkh-8(tm292)* n=76, 123, 129, 209 and *fkh-8(vlc43)* n=82, 92, 130, 139] and

*Figure 6 continued on next page*

*Figure 6 continued*

2-heptanone attraction mediated by AWC neurons [*Wild type* n=124, 129, 133; *fkh-8(tm292)* n=68, 94, 102 and *fkh-8(vlc43)* n=87, 83, 85]. Diacetyl response, mediated by AWA, is affected but not to a significant level due to high variability in the response [*Wild type* n=168, 69, 103; *fkh-8(tm292)* n=57, 85, 110 and *fkh-8(vlc43)* n=115, 107, 74]. (G to I) Attractive chemotaxis towards NaCl is unaffected in *fkh-8* mutant animals. [*Wild type* n=62, 78, 72; *fkh-8(tm292)* n=105, 116, 106 and *fkh-8(vlc43)* n=111, 52, 78]. Avoidance behavior towards toxic SDS and copper anions is significantly impaired. [Six biological replicates, 5 worms per replicate and genotype, 4 tests per worm]. Mean and standard error are represented in all graphs. See *Figure 6—figure supplement 1* for quantification of non-cilia mediated behaviors and *Figure 6—source data 1* for raw data and statistics.

The online version of this article includes the following source data and figure supplement(s) for figure 6:

**Source data 1.** Raw quantification data for behavioral analysis in *Figure 6* and *Figure 6—figure supplement 1*.

**Figure supplement 1.** FKH-8 is not required for correct display of mechanosensory behaviors mediated by non-ciliated neurons.

response to SDS, mediated by ASH and ASK ciliated neurons (*Hilliard et al., 2002*) and avoidance to copper, mediated by ASH, ASE, ADF and ADL ciliated neurons (*Guo et al., 2015*; *Sambongi et al., 1999*), were significantly reduced both in *fkh-8(vlc43)* and *fkh-8(tm292)* animals (*Figure 6H,I*, *Figure 6—source data 1*).

In summary, our battery of behavioral assays reveals FKH-8 is necessary for the correct response to a wide range of sensory stimuli (mechanical, gustatory or olfactory) that are mediated by different types of ciliated neurons (ADF, ADL, ASE, ASG, ASH, ASI, ASJ, ASK, AWB, AWC, FLP, and OLQ). Some neurons controlling affected behaviors show corresponding morphological cilia defects in *fkh-8* mutants, such as ADF, AWA, and AWB neurons. Nevertheless, we found that specific behaviors, such as attraction to NaCl or basal slowing response are sustained in *fkh-8* mutants, suggesting retained sensory functions for particular neuron types, even with gene expression or morphological cilia defects (such as CEPs).

## Mouse FOXJ1 and FOXN4, master regulators of motile ciliome, can functionally replace FKH-8

Vertebrate FKH family is composed of 49 different members divided into 16 subfamilies (*Shimeld et al., 2010*). The establishment of specific orthology relationships between FKH members is challenging among distant species (*Shimeld et al., 2010*), precluding the direct assignment of the closest vertebrate ortholog for *C. elegans* FKH-8.

To date, no vertebrate FKH TF has been shown to be involved in ciliogenesis in primary cilia cell types. Nevertheless, in several vertebrate cell types that contain motile cilia, FoxJ1 FKH TF directly activates ciliome gene expression (*Brody et al., 2000*; *Chen et al., 1998*; *Stubbs et al., 2008*; *Vij et al., 2012*; *Yu et al., 2008*). Thus, considering its role in ciliogenesis, we next wondered if mouse FOXJ1 could functionally substitute FKH-8 in *C. elegans*. We find this to be the case as FoxJ1 cDNA expression under the dopaminergic promoter *dat-1* rescues *ift-20* expression similarly to *fkh-8* cDNA (*Figure 7A-C* and *Figure 7—source data 1*). In *Xenopus*, another FKH TF, FoxN4, binds similar genomic regions to FoxJ1 and it is also required for direct ciliome gene expression in motile multiciliated cells (*Campbell et al., 2016*). We find FoxN4 expression also rescues *ift-20* expression defects in *fkh-8(vlc43)* animals. Importantly, we find that conserved functionality is not observed for any vertebrate FKH TFs as expression of mouse FoxI1, a FKH TF involved in the development of several tissues but not reported to control cilia gene expression (*Edlund et al., 2015*), does not rescue *fkh-8* mutant phenotype.

In summary, our results unravel the functional conservation between FKH-8 and specific mouse members of the FKH family, which have already been described to act together with RFX TFs in the regulation of ciliome gene expression in motile cilia cell types.

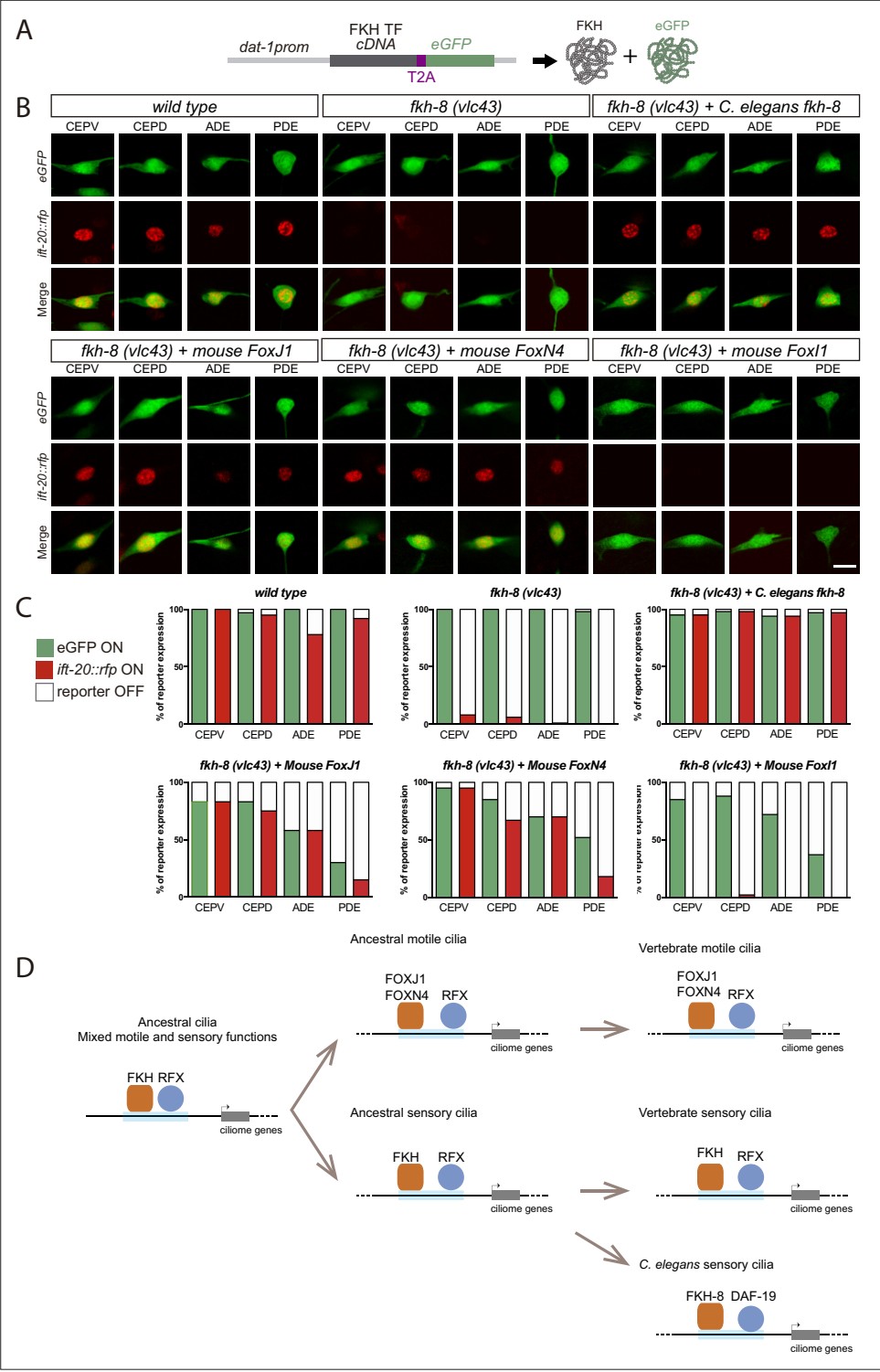

**Figure 7.** Mammalian FKH TFs with known motile cilia regulatory functions can rescue *fkh-8* mutant phenotype. (**A**) Rescue strategy: *dat-1* promoter, unaffected in *fkh-8* mutants, is used to drive FKH TF cDNA and eGFP expression specifically in the dopaminergic ciliated system. (**B**) Representative images of dopaminergic neurons expressing an integrated reporter for the core ciliome marker *ift-20* (in red) in *wild type*, *fkh-8(vlc43)* mutants and with the co-expression of different rescuing constructs. Scale bar = 5 μm. (**C**) Quantification of rescue experiments. *ift-20* reporter expression is lost from the dopaminergic neurons in *fkh-8(vlc43)* null mutants compared to *wild type* animals. Expression of FKH-8, FOXJ1, and FOXN4 but not FOXI1 is sufficient to recover *ift-20* expression in dopaminergic neurons. N=30 animals per transgenic line. See **Figure 7—source data 1** for raw data and

*Figure 7 continued on next page*

*Figure 7 continued*

similar results obtained with two additional transgenic lines per construct. (**D**) Speculative model on the evolution of ciliome gene regulatory logic. FKH and RFX TFs could have an ancestral role in the direct coregulation of ciliome genes before its functional diversification into motile and primary cilia cell types. Different RFX and FKH TF members could have evolved to regulate ciliome genes in specific cell types in different organisms. Orange squares represent FKH TFs and blue circles RFX TFs, light blue bars represent ciliome enhancers.

The online version of this article includes the following source data for figure 7:

**Source data 1.** Raw quantification data for rescuing experiments in *Figure 7*.

## Discussion

### FKH-8 acts together with DAF-19 in the direct regulation of ciliome gene expression in sensory neurons

RFX are the only TFs known to be involved in the direct coregulation of ciliome gene expression both in cell types with motile and sensory cilia. This role is conserved in nematodes, flies and vertebrates (*Choksi et al., 2014*). In this work we characterized the persistent activity of ciliome gene reporters in *daf-19/RFX* null mutants, demonstrating that, in some specific cellular contexts, DAF-19/RFX is not necessary to drive ciliome gene expression. DAF-19 is the only RFX TF in *C. elegans*; thus, persistent enhancer activity must be attributed to other TF families.

A multi-angled approach allowed us to identify FKH-8 as a key regulator of ciliome gene expression in most, if not all, sensory neurons in *C. elegans*. FHK-8 is expressed almost exclusively in all ciliated neurons and binds to upstream regions of many ciliome genes. *fkh-8* mutants show decreased levels of ciliome reporter gene expression, abnormal cilia morphology and defects in a plethora of behaviors mediated by sensory ciliated neurons. Finally, mutations in putative FKH binding sites for two ciliome reporters lead to expression defects, further supporting the direct action of FKH-8 in ciliome gene expression. Altogether, our results show that FKH-8 has a key role in regulating ciliogenesis in sensory neurons and thus represents the first identified TF in any organism that acts together with RFX in cell types with non-motile primary cilia.

In the past, the identification of direct targets of RFX TFs has been instrumental in the identification of new ciliome components, which lead to a better understanding of cilia function and the etiology of ciliopathies (*Blacque et al., 2005*; *Chen et al., 2006*; *Efimenko et al., 2005*; *Li et al., 2004*; *Schiebinger et al., 2019*). FKH-8 binds to many genes in the *C. elegans* genome, some of them with uncharacterized functions; thus, similar to RFX, a more exhaustive characterization of FKH-8 targets could be used to uncover novel components of the sensory ciliome.

### Specific DAF-19 isoforms repress *fkh-8* expression in non-ciliated neurons

Interestingly, our results show that DAF-19A and B isoforms repress (directly or indirectly) *fkh-8* expression in non-sensory neurons. Repression of alternative fates is a prevalent feature in neuronal development (*Sousa and Flames, 2022*). Repressive actions for DAF-19A/B have also been recently reported (*De Stasio et al., 2018*). Our results show that some ciliome components, such as *kap-1* can be upregulated in *daf-19* mutants even in the absence of FKH-8, thus DAF-19A/B repression of *fkh-8* might be necessary to avoid ectopic expression of some ciliome components but not others.

### Role of FKH TFs in the transcriptional regulation of ciliome genes both in motile and sensory cilia cell types

Although TFs acting together with RFX in the regulation of ciliogenesis in sensory cell types were previously unknown, RFX TFs act in concert with the FKH TF FOXJ1 in the direct regulation of ciliome genes in different vertebrate cell types with motile cilia (*Choksi et al., 2014*).

Vertebrate sensory ciliogenesis has been reported to be unaffected in FoxJ1 loss of function mutants (*Choksi et al., 2014*); thus, FoxJ1 role as a master regulator of ciliogenesis seems restricted to motile ciliary cell types. Interestingly, a recent report indicates that both zebrafish and mouse FoxJ1 mutants display ciliary defects in Olfactory Sensory Neurons, which display an atypical 9+2 sensory cillium (*Rayamajhi et al., 2023*). In *Xenopus*, FoxN4 binds similar genomic regions to FoxJ1 and it

is also required for motile ciliome gene expression (*Campbell et al., 2016*). We find both FOXJ1 and FOXN4, but not FOXI1, which has not been described to be involved in ciliogenesis, are able to functionally substitute FKH-8. In *C. elegans*, another FKH TF, FKH-2, is a downstream target of DAF-19 that controls expression of some ciliome components and cilium morphogenesis specifically in AWB neurons (*Mukhopadhyay et al., 2007*). Finally, in *Drosophila melanogaster*, fd3F FKH transcription factor directly regulates expression of several ciliome genes in the chordotonal neurons that contain atypical cilia with both sensory and motile functions (*Newton et al., 2012*). Altogether, this data suggests that specific FKH TFs might have the capacity to act as direct ciliome regulators, independently of being expressed in motile or sensory cilia cell types.

Importantly, vertebrate sensory ciliogenesis is unaffected in FoxJ1 loss of function mutants (*Choksi et al., 2014*); thus, FoxJ1 role as a master regulator of ciliogenesis is restricted to motile ciliary cell types. In *Xenopus*, FoxN4 binds similar genomic regions to FoxJ1 and it is also required for motile ciliome gene expression (*Campbell et al., 2016*). We find both FOXJ1 and FOXN4, but not FOXI1, which has not been described to be involved in ciliogenesis, are able to functionally substitute FKH-8. In *C. elegans*, another FKH TF, FKH-2, is a downstream target of DAF-19 that controls expression of some ciliome components and cilium morphogenesis specifically in AWB neurons (*Mukhopadhyay et al., 2007*). Altogether, this data suggests that specific FKH TFs might have the capacity to act as direct ciliome regulators, independently of being expressed in motile or sensory cilia cell types.

## FKH-8 and DAF-19 show synergistic actions

FKH-8 bound regions are enriched for X-box/RFX sites, FKH-8 and DAF-19 physically interact and double mutant analysis shows synergistic effects between *daf-19/RFX* and *fkh-8*, altogether these data suggest DAF-19 and FKH-8 cooperate in the regulation of a common set of regulatory regions. Cooperativity between FKH and RFX has been reported in vertebrates, in motile multiciliated cells of *Xenopus* larval skin FOXJ1 binding to ciliome gene promoters depends on the presence of RFX2 (*Quigley and Kintner, 2017*). In addition, in human airway multiciliated epithelial cell, RFX3 and FOXJ1 act synergistically in the activation of ciliome genes (*Didon et al., 2013*).

Importantly, cooperative actions of DAF-19 and FKH-8 seem prevalent for core ciliome components while several sub-type specific ciliome genes are known targets for neuron-type-specific terminal selectors (*Etchberger et al., 2007*; *Flames and Hobert, 2009*; *Zhang et al., 2014*).

## Evolution of cilia subtype specialization and ciliome regulatory logic

Ancestral cilium present in the last common eukaryotic ancestor has been proposed to combine motile and sensory functions (*Mitchell, 2017*). RFX role regulating ciliome expression predates the emergence of metazoans, where major cell type diversification has occurred (*Chu et al., 2010*; *Piasecki et al., 2010*). FoxJ and FoxN constitute the most ancient FKH sub-families, present in choanoflagelate *Monosiga brevicolis*, while FoxI subfamily is only present in bilaterians (*Shimeld et al., 2010*). Moreover, the ability of RFX and FKH TFs to bind similar genomic regions is not limited to metazoans and it is also present in fungi. For example, in *Schizosaccharomyces pombe,* which lacks cilia and ciliome genes, Fkh2 FKH TF and Sak1 RFX TF bind the same regulatory regions to control cell cycle gene expression (*Garg et al., 2015*), suggesting that the joint actions for these TFs could be present before the split of fungi and metazoans. Alternatively, RFX and FKH TFs might have an inherent ability to cooperate that could explain convergent evolution of these TFs in ciliome regulation both in sensory and motile cilia cell types (*Sorrells et al., 2018*).

In light of these data, we hypothesize that RFX and FKH role as co-regulators of ciliome gene expression could precede the emergence of cilia division of labor and the specialization of motile and sensory cilium in different cell types (*Figure 7D*).

## Role of FKH TFs in ciliome regulation of primary cilia cell types

Regardless of the evolutionary history of events underlying RFX and FKH functions as direct regulators of ciliome gene expression, our results raise the possibility that, in vertebrates, yet unidentified FKH TFs could act together with RFX in the regulation of ciliome gene expression in sensory ciliated cell types (*Figure 7D*). The establishment of specific orthology relationships between FKH members among distant species is challenging (*Larroux et al., 2008*; *Shimeld et al., 2010*) precluding the direct assignment of the closest vertebrate ortholog for *C. elegans* FKH-8. In addition, functional paralog

substitutions among TFs of the same family have been described to occur in evolution (*Tarashansky et al., 2021*). Importantly, although FoxJ1 and FoxN4 mutants do not show major ciliome gene expression defects in non-motile ciliated cell types (*Brody et al., 2000*; *Campbell et al., 2016*; *Chen et al., 1998*; *Stubbs et al., 2008*; *Yu et al., 2008*) a recent report describes ciliary defects in vertebrate Olfactory Sensory Neurons in Foxj1 mutants (*Rayamajhi et al., 2023*). Other members of FoxJ and FoxN subfamilies are broadly expressed in mouse neurons, which all display primary cilia (*Zeisel et al., 2018*). It will be important, in future studies, to determine if additional FoxJ and FoxN TFs can rescue *fkh-8* expression defects in *C. elegans* and if they display similar roles in mammals. These studies could also help better characterize the functional meaning of non-coding mutations associated to orphan ciliopathies.

## Methods
### Materials availability
Newly generated strains are listed in *Supplementary file 1* and accessible through *Caenorhabditis* Genetics Center (CGC). Plasmids are available upon request.

### *C. elegans* strains and genetics
*C. elegans* culture and genetics were performed as previously described (*Brenner, 1974*). Strains used in this study are listed in *Supplementary file 1*.

### Mutant strain genotyping
Mutant strains used in this study are listed in *Supplementary file 1*. Deletion alleles were genotyped by PCR. Presence of *daf-19(m86)* allele was determined by visual inspection of the dye-filling defective phenotype of homozygous mutants. Presence of *daf-12(sa204)* allele was ensured through a double-cross strategy, crossing of F1 males with original *daf-12(sa204)X* mutants. Strains carrying point mutations were genotyped by sequencing. Genotyping primers are included in *Supplementary file 1*.

### DiD staining
Lipophilic dye filling assays were performed with the 1,1′-dioctadecyl-3,3,3′,3′-tetramethylindodicarbocyanine, 4-chlorobenzenesulfonate salt (DiD) (Thermofisher, #D7757). DiD staining solution was freshly prepared prior to every assay as a 1:200 dilution of the DiD stock solution [2 mg/mL dilution in N,N-dimethyl formamide (Sigma, #D4551)] in M9 1 X buffer. Animals were transferred into 1.5 mL tubes containing 200 µL of the DiD staining solution and incubated (wrapped in aluminium foil) for 2 hr at room temperature in an orbital shaker in a horizontal position. Animals were collected with a glass Pasteur and transferred to fresh NGM plates. Robust identification of the ASK, ADL, ASI, AWB, ASH, ASJ, PHA and PHB ciliated neurons was achieved through this method.

### Generation of *C. elegans* transgenic lines
Fluorescent reporters for ciliome genes were generated through fusion PCR (*Hobert, 2002*). To facilitate identification and scoring of reporter-expressing cells, GFP was tagged to the cell's nucleus employing a modified sequence of the classical SV40 large T antigen nuclear localizing signal (NLS) (*Kalderon et al., 1984*). Regulatory sequences were amplified with custom oligonucleotides from N2 genomic DNA preparations. An independent PCR was used to amplify the 2xNLS::GFP::*unc-54* 3′UTR fragment from an NLS version of the pPD95·75 plasmid (pNF400). Successfully fused PCR products were purified using the QIAquick PCR Purification Kit (QIAGEN, #28106) and resuspended in nuclease-free water (Sigma, #W4502).

Mutated versions for the *xbx-1* and *ift-20* promoters were generated as PCR products by introducing the desired mutation of putative FKH sites within the corresponding custom primers. Putative FKH sites were identified through the single sequence scan tool from the CIS-BP website (*Weirauch et al., 2014*). Mutation criteria accounted for the nature of the nitrogenous bases and the number of hydrogen bonds they could form; thus, A was mutated to C and G was mutated to T (and vice versa). Mutated sequences were checked to discard the generation of new TF binding site motifs using both the motif scan tool of the CIS-BP database and the Tomtom tool (*Gupta et al., 2007*) from the MEME

Suite website. When designed mutations created potential new TF binding sites manual punctual mutations were applied to disrupt those potential sites.

To generate FKH-8 rescuing plasmids, constructs containing the cDNA of the corresponding FKH TF fused to the self-cleaving peptide T2A (*Ahier and Jarriault, 2014*) and the eGFP cDNA from the pPD95.75 plasmid were created. Such constructs were then cloned under the control of the dopaminergic *dat-1* promoter between the KpnI/XhoI sites of the pPD95.75 backbone vector. *fkh-8* cDNA sequence was synthetically generated (Biomatik). Murine FKH members were obtained as Dharmacon clones (FoxJ1: MMM1013-202732974, FoxN4: MMM1013-211694291, FoxI1: MMM1013-202763055).

Simple-array transgenic lines were generated by intragonadal microinjection into strains of the appropriated genotype. The injection mix was composed by 50 ng/µL of a given purified fusion PCR or a rescuing plasmid plus 100 ng/µL of the pFR4 plasmid, *rol-6(su1006),* as a co-marker (*Mello et al., 1991*).

## Generation of *C. elegans* mutations

Whole deletion of the *fkh-8* locus was performed through a co-CRISPR strategy (*Kim et al., 2014*) using *dpy-10(cn64)* as conversion marker (*Arribere et al., 2014*). Custom CRISPR RNAs (crRNAs) were ordered (IDT, Alt-R CRISPR-Cas9 crRNA XT) targeting both sides of the desired deletion of *fkh-8* and at the 5' of the *dpy-10* site of mutation. Single-stranded oligodeoxynucleotide (ssODNs) of approximately 100 base pairs overlapping each side of the genetic modifications were also ordered (IDT) and used as donor templates to achieve homology-directed repair. Cas9 nuclease (IDT, #1081058) and the universal trans-activating crRNA (tracrRNA) needed to initiate enzymatic activity (IDT, #1072532) were used. co-CRISPR injections were performed on young adult hermaphrodites expressing the reporter *otIs395(ift-20::NLS::tagRFP)III*. Microinjection mix was freshly prepared with all 3 crRNAs plus the tracrRNA, ssODNs and Cas9 nuclease. Ribonucleoprotein complex formation was achieved by incubating this mix for 10 min at 37°C. Before use, the final mix was incubated on ice for 30 min. *daf-19(of5)* was generated by CRISPR deletion of 9 bp that include the ATG for the *daf-19 a* and *daf-19 b* isoforms and inserting a guanine nucleotide N2 sequence: GGCAGAGAAGAAAGT**CATGACCA** ATGAGGAGCC; of5 sequence GGCAGAGAAGAAAG**g**ATGAGGAGCC. Knock-in strain PHX6528 *[osm-5(syb6528), osm-5::SL2::GFP::H2B]* was generated by SunyBiotech's CRISPR services. All custom primer sequences and concentrations used for the generation of the aforementioned strains are included in the *Supplementary file 1*.

## Behavioral assays

Unless otherwise stated, all mechano- and chemosensory assays were performed over small-scale synchronized populations of young adult hermaphrodites.

Nose touch tests were performed as previously described (*Kaplan and Horvitz, 1993*). Ten minutes before the assay, young adult hermaphrodites were transferred to non-seeded NGM agar plates and nose touch responses were elicited by causing a nose-on collision placing an eyelash attached to a pipette tip in the path of an animal moving forward. With brief modifications from *Brockie et al., 2001*, five consecutive nose touch trials were scored for each worm.

Both gentle and harsh touch mechanosensory tests were performed as previously described (*Chalfie et al., 1985*). Briefly, gentle touch assays were performed by alternatively stroking the animal just behind the pharynx and just before the anus with an eyebrow hair attached to a pipette tip for a total amount of 10 strokes (*Hobert et al., 1999*). Harsh touch assays were also performed by stroking the worms across the posterior half of their bodies in a top-down manner with a platinum wire. Each worm was tested five times with a 2 min interval between each trial (*Li et al., 2011*).

For all aforementioned mechanosensory assays, escape responses of trailed animals were recorded and a population response index (RI) was calculated for every replica as: RI = total number of escape responses / total amount of strokes Chemotaxis towards diacetyl, 2-heptanone, NaCl and 2-nonanone were performed over three times freshly washed worms with 1 mL of filtered, autoclaved CTX solution, aspirating the supernatant to a final volume of approximately 100 µL. Two µL of this worm-containing solution were placed at the proper place of the assay plates. During the assays, worms were allowed to freely crawl across the plates for 60 min at room temperature and then stored at 4 °C until the next day when worms' positions were scored and behavioral indexes were calculated.

With few modifications, volatile diacetyl attraction assay was performed as described by *Margie et al., 2013*. A four-quadrant paradigm drawn at the base of non-seeded NGM agar plates was used, adding a 1 cm circular central area that worms had to trespass to be scored. Stock diacetyl (Sigma-Aldrich, #803528) test solution was prepared as a 0.5% V/V mix in absolute ethanol (Scharlau, #ET00101000). Absolute ethanol was used as control solution. Immediately after the worms were plated, 2 µL of a mix combining equal volumes of diacetyl stock solution and sodium azide 1 M were pipetted onto the 2 test sites (T) of the agar plate. Same procedure was then performed for the 2 control sites (C). Chemotaxis index (CI) was then calculated as: CI = (worms in (T1 +T2) - worms in (C1 +C2)) / total scored worms.

Chemotaxis assay towards 2-heptanone was performed as previously reported (*Zhang et al., 2016*). A two-halves paradigm was used, adding the threshold distance by *Margie et al., 2013* to prevent immobile worms from skewing the data. 2-Heptanone (Sigma-Aldrich, #W254401) test solution was prepared as a 1:10 V/V mix in ethanol absolute. Ethanol was used as control solution. Immediately after the worms were plated, 3 µL of a mix combining equal volumes of 2-heptanone stock solution and sodium azide 1 M were pipetted onto the test site (T) of the agar plate. Same procedure was follow to the control site (C). CI was calculated: CI = (worms in T - worms in C) / total scored worms.

Chemotaxis toward NaCl was also performed of a two-halves paradigm. Radial gradients of either test or control solutions were created prior to worm loading as originally stated (Ward 1973). Following (*Frøkjaer-Jensen et al., 2008*), 10 µL of NaCl (Sigma, #S3014-1KG) 2.5 M (dissolved in double distilled water (ddH2O)) or ddH2O itself were respectively pipetted onto the agar surface at T and C spots and allowed to diffuse for 12–14 hr at room temperature. To increase steepness of the gradients, 4 µL of NaCl 2.5 M or ddH$_2$O solutions were additionally added to the T and C spots respectively 4 hr prior to the chemotaxis assay. Chemotaxis indexes for two-halves paradigm assays were calculated as: CI = (worms in T - worms in C) / total scored worms.

Avoidance assay towards 2-nonanone was performed as previously reported (*Troemel et al., 1997*). Briefly, six equal sectors labeled as A, B, C, D, E, and F were drawn on the base of squared plates (90x15 mm, Simport, # 11690950) containing 15 mL of standard NGM agar. Stock 2-nonanone (Sigma-Aldrich, #W278550) test solution was prepared as a 1:10 V/V mix in absolute ethanol. Ethanol was used as control solution. Immediately after the worms were plated on the centre of the plate, 2 µL of a mix combining equal volumes of 2-nonanone stock solution and sodium azide 1 M were pipetted onto two spots within peripheral test sector A. Same procedure was then performed for the ethanol control sites within sector opposite peripheral control sector F. Population avoidance index (AI) was calculated as: AI = (worms in (A+B) - worms in (E+F)) / total amount of worms.

Avoidance responses to water-soluble compounds were evaluated using the drop test as previously described *Hilliard et al., 2004* following a few modifications. Well-fed synchronized young adult hermaphrodites were washed three times with M13 buffer, 5 animals were then placed on unseeded NGM agar plates and allowed to rest for 10 min. Two test solutions were assayed: 0.1% W/V sodium dodecyl sulfate (SDS) (Sigma, #L3771-100G) and 0.1 mM CuSO4 pentahydrate (Merck, #1027901000), both dissolved in the M13 buffer that acted as control solution. Each animal was tested first with 4 single drops of the control solution and then with 4 single drops of the testing solution, allowing for 2 min of recovery between each stimulus. Avoidance response was scored within 4 s after substance delivery. Population avoidance index (AI) per genotype and replica was calculated as: AI = number of responses / total amount of drops.

Dauer induction was performed using filtered liquid culture obtained from *wild type* worms grown at 7 worms/µl for 4 days. Briefly, 300 µl of pheromone containing extracts or control extracts (culture media alone) were added to 60 mm OP50-seeded NGM plates. After drying, 10 gravid worms were added and allowed to lay eggs for 18 hr and then removed from the plates. Seventy-two h later, resulting P0 worms were scored and percentage of dauer animals determined for each condition. Dauer induction was carried at 27 °C in four independent experiments performed in parallel with wild type and *fkh-8(vlc43)* mutant worms.

Basal slowing response was performed with few modifications as previously reported (*Sawin et al., 2000*). In this case, 60 mm NGM plates in which HB101 was seeded in only one half of the plate were used. Briefly, well-feed worms were 3 times washed with 1 mL of filtered, autoclaved CTX solution, supernatant aspirated to a final volume of approximately 200 µL and 2 µL of this worm-containing solution (with no less than 10 animals) was placed at the non-seed part of pre-warmed assay plates.

Free movement of the worms across the plates was recorded capturing 30 frames per second. Body beds per 20 s intervals were counted from same worms moving on agar and crawling across the bacterial lawn.

Sample size, tested genotypes, number of animals and number of replicates performed per assay are detailed in the corresponding figure legends and in **source data** Files. All strains used for these behavioral studies are listed in **Supplementary file 1**.

## Co-immunoprecipitation experiments

Human optimized sequences for FLAG:DAF-19 and HA:FKH-8 were cloned into PCDNA3.1 plasmid. A 10 cm plate of HEK293 cells (Human 293T, ATCC CRL-3216, authenticated by microsatellite amplification, Secugen, tested micoplasm negative) was transfected with 5 µg of each plasmid. At 48 hr post-transfection cells were washed three times with ice-cold PBS and lysed for 10 min on ice with 200 µl of cytoplasmic fractionation buffer containing 10 mM Hepes pH 7.9; 10 mM KCl; 1.5 mM MgCl2; 0.34 M sucrose, 10%glycerol; 1 mM DTT; 5 ug/ml protease inhibitor; 0.1 mM PSMF and 0.1% Triton X-100 from a 10% stock. Cells were centrifuged at 3500 rpm for 4 min at 4 °C. The supernatant (cytoplasmic fraction), was collected and clarified by centrifugation (15 min 14,000 rpm at 4 °C).

The pellet (nuclei fraction) was lysed for 30 min on ice with 100 µl of nuclear fractionation buffer comprising 10 mM Hepes pH 7.9, 3 mM EDTA, 0.5 mM NaF, 0.2 mM EGTA, 1 mM DTT; 5 ug/ml protease inhibitor and 0.1 mM PSMF. The fraction was homogenized, incubated for 10 min on ice and centrifuged at 4000 rpm for 4 min at 4 °C. The supernatant (soluble nuclear fraction), was collected. The pellet (chromatin fraction) was resuspended with 300 µl of nuclear fractionation buffer and was sonicated for 15–20 s (25% amplitude).

Total protein concentration of soluble cytoplasmic fraction, soluble nuclear fraction and chromatin fraction were used for coimmunoprecipitation assays. The association between DAF-19 and FKH-8 was analyzed by coimmunoprecipitation using anti-HA magnetic beads (bimake.com). Cytoplasmic and nuclear fractions were incubated with 10 µl of magnetic beads overnight at 4 °C with rotation. The beads were separated by a magnetic separator and washed three times in 500 µl TNE buffer 0.1% Triton X-100. Finally, for reducing SDS-PAGE analysis, 30 µl 2xLaemmli buffer (with 2% ß-MeOH) was added and the samples were boiled for 5 min at 96 ° C. Anti-mouse Light chain-specific secondary antibodies (Jackson Immunoresearch ref: 115-655-174) were used for detection since the heavy chain from the immunoprecipitation would mask the FKH-8 signal [anti-HA antibody (Biolegend 901501), anti-FLAG antibody (Sigma, F1804)]. Cytoplasmic and nuclear fractions were controlled using antibodies against MEK2 (Becton Dickinson, 610235), marker for cytoplasmic fraction and histone H3, marker for chromatin associated fraction (Abcam ab1791).

## Microscopy

For scoring and image acquisition, worms were anesthetized with a drop of 0.5 M sodium azide (Sigma, #26628-22-8) on 4% agarose pads (diluted in distilled water) placed over a regular microscope glass slide (Rogo Sampaic, #11854782). These preparations were sealed with a 24x60 mm coverslip (RS France, #BPD025) and animals were then conveniently examined.

Scoring of ciliome features was performed observing the animals on a Zeiss Axioplan 2 microscope using a 63 X objective. Assessment of fluorescence signal on PDE and Phasmid regions was performed *de visu*. To appropriately assess number of cells in the head, optical sections containing the volume of reporter-positive neurons in the head of the animals were acquired at 1 µm intervals and images were manually scored using FIJI (**Schindelin et al., 2012**). Reporters used in the FKH cis-mutational analyses (both *wild type* and mutated forms) were scored *de visu* as the low intensity and fast bleaching in their signals precluded us from taking pictures.

Fluorescence intensity levels from the endogenous *osm-5::SL2::GFP::H2B* reporter strain were measured on young adult animals grown at 25 °C. All images were acquired with a TCS-SP8 Leica Microsystems confocal microscope using a 63 X objective on animals immobilized as previously described. Image acquisition was optimized considering the appropriate no saturating conditions for the *wild type* background. To avoid a possible bias induced by the volume of the worm's body, *dat-1::cherry* reporter was used to select a single section for the PDE, CEPV, CEPD and ADE nuclei and was used for *osm-5::GFP* fluorescence quantification. The contour of each nucleus was delineated and fluorescence intensity quantified using FIJI. For each cell, corrected total cell fluorescence was

calculated as follows: Integrated Density – (Area of selected cell X Mean fluorescence of background reading).

For cilia morphology assessment, 0.5 M sodium azide was used as an immobilization agent image acquisition was performed with a TCS-SP8 Leica Microsystems confocal microscope using a 63 X objective. The following conditions of optical sections (μm) were used: CEP: 0.4 μm; ADF: 0.2; AWB: 0.24; AWC: 0.3. Retrieved images were z-projected at maximum intensity (Leica LAS X LS) and linear adjustment for brightness and contrast was performed prior to cilia length quantification (N≥32 cilia per neuron type; FIJI). AWA analysis was performed from images acquired from dorsoventrally positioned animals (N=7) in which both cilia were levelled and depth of arborisation was estimated from the volume containing all the optical sections (0.3 μm) in which fluorescence signal was observed. Ciliary morphology was also measured on animals immobilized as previously reported (*Niwa, 2017*) using 10% (w/v) agarose pads and 2.5% solids (w/v) aqueous suspension of polystyrene microspheres with 100 nm of diameter (Polysciences, #00876–15). Optical sections containing the volume of reporter-positive cilia were acquired at 0.3 μm intervals. Images were z-projected at maximum intensity (Leica LAS X LS) and linear adjustment for brightness and contrast was performed prior to ciliary length quantification (N≥30 cilia per neuron type) (FIJI).

All micrographs presented in this paper were acquired with a TCS-SP8 Leica Microsystems confocal microscope using a 63 X objective and appropriate zooming conditions.

## Statistical analyses

Statistical significance for the mean number of reporter-positive neurons in whole animals among different genetic backgrounds was assessed by two-tailed t-test. Inbuilt Excel functions F.TEST and T.TEST were used and obtained p-values were adjusted through Bonferroni correction accounting for all possible pairwise comparisons in each experiment.

To increase for statistical power, statistical significance for the mean number of reporter-positive neurons in the five distinct anatomical regions containing ciliated neurons among different genetic backgrounds was assessed by one-tailed t-test. Obtained p-values were then adjusted through the Benjamini-Hochberg procedure setting α level at 0.05. This same procedure was used to assess for statistical significance within the dauer induction experiments.

Unless otherwise stated, same two-tailed t-test procedure was followed in the assessment of statistical significance in behavioral experiment. Behavioral responses were ultimately analysed through the corresponding indexes ranging from 0 to 1 (or to –1–0 when avoidance responses were assayed). For each type of assay, a population-based mean index was calculated per replica and a final response index was then obtained as the mean of all replicas' means. Prior to hypothesis testing, the Shapiro-Wilk test (*Shapiro and Wilk, 1965*) was used to address for the normality of these final indexes.

Assessment of synergistic effects between *fkh-8* and *daf-19* was performed under the multiplicative model (*Wagner, 2015*). Briefly, average number of reporter-expressing neurons found in the whole animals for each genetic background was transformed into the corresponding fold change related to the observed mean value in the *daf-12* single mutant. Next, expected values for the fold change corresponding to triple *daf-12; daf-19, fkh-8* mutants were calculated as the product of the mean observed values for the double *daf-12; daf-19* and daf-12; *fkh-8* mutant strains. To calculate the associated error for this indirect measure, propagation error was used (sum of each standard deviation between the corresponding mean from each observed value in each double mutant multiplied by the value of the expected value). Statistical significance between observed and expected values was then assessed through a one-sample t test.

For the assessment of statistical significance in rescue experiments, data was categorically classified as 'on' or 'off' and the significance of the association was examined using the two-tailed Fisher's exact test. No further multiple testing correction was performed, as *fkh-8* null mutants were exclusively compared to wild type worms whereas each rescued line was exclusively compared against the *fkh-8* null mutants.

## Bioinformatics analysis

Ciliome gene list was assembled including genes associated with cilium-related terms from the Gene Ontology using AmiGO (*Carbon et al., 2009*), known ciliome genes with functional X-boxes (*Burghoorn et al., 2012*) and genes whose expression in ciliated neurons was reported in the

WormBase. Transcription factors were deliberately excluded from this list. A further curation process was performed through a bibliographic research (see *Figure 1—source data 1* for complete ciliome gene list).

For each isoform of the final 163 genes composing the ciliome gene list, putative regulatory sequences were retrieved from the Ensembl BioMart site (*Kinsella et al., 2011*) spanning 700 base pairs in length upstream of their translational start sites. These sequences were used to feed the RSAT oligo-analysis tool as previously described (*Defrance et al., 2008*; *Turatsinze et al., 2008*), using as a background model the in-tool genome of *C. elegans* and overall default options. Retrieved matrices were then compared both against the CIS-BP 1.02 (*Weirauch et al., 2014*) and the JASPAR core non-redundant 2018 (*Khan et al., 2018*) databases using the TomTom (*Gupta et al., 2007*) tool from the MEME suite (*Bailey et al., 2009*). Four different matrices of different lengths all matching RFX binding sites (X-boxes) were retrieved through this method. Overlapping of these matrices over the putative regulatory sequences from the ciliome genes was used to defined X-box regions whose coordinates were assessed for the ce10 version of the *C. elegans* genome.

Identification of candidate transcription factors with enriched expression in ciliated neurons was performed through the on-line tool GExplore$_{1.4}$ (*Hutter and Suh, 2016*), employing the sci-RNA-seq dataset by *Cao et al., 2017*. A fivefold enrichment ratio and a false detection rate of 0.001 were used.

Expression pattern data in each ciliated neuron type for candidate transcription factors at the fourth larval stage were retrieved from the *C. elegans* Neuronal Gene Expression Network (CeNGEN; *Taylor et al., 2021*), whose results are freely accessible through the on-line tool SCeNGEA. Unfiltered data was used.

ChIP-seq data from *C. elegans* TFs were retrieved from the ENCODE portal website (*Davis et al., 2018*) (time of consulting: January the 10th, 2019). Peak annotation was carried out employing the ChIPseeker package (*Yu et al., 2015*), setting parameters as following: annotatePeak(gr1, tssRegion = c(–2000, 1000), level = lev, TxDb = annoData, overlap="TSS"). ENCODE accession numbers for all datasets used in this analysis are listed in *Figure 1—source data 2*. *fkh-8* ChIP-seq bed narrowPeak file (ENCODE accession: ENCFF653QKE) was used as input file for the web-based analysis tool ChIP-seek (*Chen et al., 2014*). For de novo motif discovery, resulting fasta file with annotated peaks was then used to feed the RSAT peak-motifs tool as previously described (*Thomas-Chollier et al., 2012a*; *Thomas-Chollier et al., 2012b*), setting the number of motifs per algorithm at 10 and using all 4 available discovery algorithms with overall default options.

For gene onthology, genes associated to FKH-8 ChIP-seq peaks where analysed through the on-line tool WormEnricher (*Kuleshov et al., 2016*).

Gene expression correlation between TFs and genes of interest were calculated using embryonic sc-RNA-seq data (*Packer et al., 2019*). Genes of interest were categorized into four categories: (1) core ciliome genes, (2) subtype-specific ciliome genes (both extracted from our ciliome list), (3) panneuronal genes (*Stefanakis et al., 2015*) and (4) ubiquitously expressed genes (*Packer et al., 2019*). In addition to fkh-8 and daf-19, the proneural TF factor hlh-14 was added as control TF not related to ciliogenesis. For all 10,775 ciliated cells present in the dataset, correlation index (R) between the expression levels for each gene and the TF was calculated. R data for each gene category are represented in the graph (See *Figure 2—source data 2* for R values).

Presence of RFX/*daf-19* binding motifs within the FKH TFs ChIP-seq peak sequences was performed with the on-line tool Centrimo (*Bailey and Machanick, 2012*) from the MEME suite. Sequences 420 base pairs in length spanning 210 base pairs from the centre of each peak were extracted to prevent Centrimo from discarding sequences due to uneven sequence length within and among the different ChIP-seq datasets. This consensus length was used considering the average sequence length of FKH-8 ChIP-seq peaks. ENCODE accession numbers for all datasets used in this analysis are listed in *Figure 2—source data 2*.

Visualization and analysis of ChIP-seq and RNA-seq files were performed with the Integrative Genomics Viewer (IGV) software (*Robinson et al., 2011*).

## Acknowledgements

We thank CGC (P40 OD010440) for providing strains. Dr Laura Chirivella, Noemi Daroqui, Anna Roig and Elia García for technical help. Erick Sousa for providing bioinformatics assistance. Ioannis Segos and Barbara Conradt for sharing the immobilization protocol, Ethel Queralt for advice on the Co-IP

experiments and Ines Carrera and Elisa Martí for comments on the manuscript. Funding: This work was supported by European Research Council (StG2011- 281920 and COG-101002203), Ministerio de Ciencia e Innovación (SAF2017-84790-R and PID2020-115635RB-I00) and Generalitat Valenciana (PROMETEO/2018/055).

# Additional information

## Funding

| Funder | Grant reference number | Author |
| --- | --- | --- |
| HORIZON EUROPE European Research Council | ERC-2020-COG-101002203(NEUROCODE) | Rebeca Brocal-Ruiz Ainara Esteve-Serrano Carlos Mora-Martínez Nuria Flames |
| Ministerio de Ciencia e Innovación | BES-2015-072799 | Rebeca Brocal-Ruiz |
| Ministerio de Ciencia e Innovación | PID2020-115635RB-I00 | Rebeca Brocal-Ruiz Ainara Esteve-Serrano Carlos Mora-Martínez Nuria Flames |
| European Research Council | ERC-2011-StG_20101109 | Rebeca Brocal-Ruiz Ainara Esteve-Serrano Carlos Mora-Martínez Nuria Flames |
| Ministerio de Ciencia e Innovación | SAF2017-84790-R | Nuria Flames |
| European Research Council | COG-101002203 | Nuria Flames |
| Generalitat Valenciana | PROMETEO/2018/055 | Nuria Flames Marçal Vilar |

The funders had no role in study design, data collection and interpretation, or the decision to submit the work for publication.

## Author contributions

Rebeca Brocal-Ruiz, Formal analysis, Investigation, Visualization, Writing - review and editing; Ainara Esteve-Serrano, Formal analysis, Investigation; Carlos Mora-Martínez, Data curation, Methodology; Maria Luisa Franco-Rivadeneira, Investigation; Peter Swoboda, Resources, Writing - review and editing; Juan J Tena, Formal analysis; Marçal Vilar, Investigation, Methodology; Nuria Flames, Conceptualization, Supervision, Funding acquisition, Visualization, Writing - original draft, Project administration, Writing - review and editing

## Author ORCIDs

Rebeca Brocal-Ruiz ⓘ http://orcid.org/0000-0003-3375-942X
Peter Swoboda ⓘ http://orcid.org/0000-0001-6416-8572
Marçal Vilar ⓘ http://orcid.org/0000-0002-9376-6544
Nuria Flames ⓘ http://orcid.org/0000-0003-0961-0609

## Decision letter and Author response

Decision letter https://doi.org/10.7554/eLife.89702.sa1
Author response https://doi.org/10.7554/eLife.89702.sa2

# Additional files

## Supplementary files

- Supplementary file 1. List of reagents: Strains, plasmids and primers.
- MDAR checklist

## Data availability

All data generated or analysed during this study are included in the manuscript and supporting file; Supplementary File 1 includes strains, plasmids and primers.

The following previously published datasets were used:

| Author(s) | Year | Dataset title | Dataset URL | Database and Identifier |
|---|---|---|---|---|
| Davis CA, Hitz BC, Sloan CA, Chan ET | 2018 | The 1168 Encyclopedia of DNA elements (ENCODE): Data portal update | https://www.encodeproject.org/ | ENCODE, ENCFF549ZSK |
| Packer JS, Zhu Q, Huynh C, Sivaramakrishnan P, Preston E, Dueck H, Stefanik D, Tan K, Trapnell C, Kim J, Waterston RH, Murray JI | 2019 | A lineage-resolved molecular atlas of C. elegans embryogenesis at single cell resolution | https://www.ncbi.nlm.nih.gov/geo/query/acc.cgi?acc=GSE126954 | NCBI Gene Expression Omnibus, GSE126954 |
| Taylor SR, Miller DM | 2019 | Molecular topography of an entire nervous system | https://www.ncbi.nlm.nih.gov/geo/query/acc.cgi?acc=GSE136049 | NCBI Gene Expression Omnibus, GSE136049 |

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
