## [Editor Report]

Sensory cilia are highly specialized organelles whose development and function requires complex machinery. In this important work, the authors use a convincing set of genetic, bioinformatic, and biochemical approaches in *C. elegans* to demonstrate that the forkhead transcription factor FKH-8 acts cooperatively with the RFX factor DAF-19 to activate the expression of many cilium genes. These findings indicate that forkhead factors have an ancient, conserved role in the development of sensory cilia, making this paper of interest to a variety of developmental and cell biologists.

---

## [Decision Letter]

**Decision letter after peer review:**

[Editors’ note: the authors submitted for reconsideration following the decision after peer review. What follows is the decision letter after the first round of review.]

Thank you for submitting the paper "Forkhead transcription factor FKH-8 is a master regulator of primary cilia in *C. elegans*" for consideration by *eLife*. Your article has been reviewed by 2 peer reviewers, , including Douglas Portman as the Reviewing Editor and Reviewer #1, and the evaluation has been overseen by a Senior Editor.

We are sorry to say that, after consultation with the reviewers, we have decided that this work will not be considered further for publication by *eLife*. However, if you are able to carry out the significant additional work that would be required to address the concerns below, we would be willing to consider a significantly revised manuscript as a new submission.

As you will see, both reviewers find the question you address, as well as the identification of fkh-8, to be of interest. However, there are multiple significant concerns with the work as it stands. In particular, Reviewer #2 raises important concerns about calling fhk-8 a "master regulator," given its relatively weak mutant phenotypes, as well as concerns about a lack of clarity regarding the nature of the functional relationship between fkh-8 and daf-19. (Although it is not specifically noted in either of the reviews below, showing that ectopic expression of fkh-8, or perhaps fkh-8 and daf-19 together, is sufficient to activate cilium genes in non-ciliated neurons might help address some of these issues.) Related to this are concerns, described below, with the methods used to detect and quantify cilium gene expression. There is also an important concern about using language rigorously -- this arises, for example, by repeatedly discussing the binding of FKH-8 to particular regions without any direct demonstration of this, and by using terms like 'master regulator' and 'synergy' without strong support. We hope that you will find the detailed comments of the reviewers listed below helpful.

*Reviewer #1 (Recommendations for the authors):*

In this paper, Brocal-Ruiz et al. set out to identify factors that work together with the RFX factor DAF-19 in the development of sensory cilia in *C. elegans*. Using an informatic approach, they identify the Forkhead-family transcription factor FKH-8 as a candidate for this role. Their findings largely support the idea that fkh-8 is expressed broadly in sensory cilia, that it acts cell-autonomously and directly to activate the expression of many cilium-specific genes, and that full expression of these genes may depend on synergy between DAF-19 and FKH-8. Consistent with this, animals lacking fkh-8 have defects in cilium morphology and in many sensory behaviors that require the function of ciliated neurons. Interestingly, they provide evidence that some vertebrate Forkhead-family factors can substitute for fkh-8 in *C. elegans*, suggesting that cooperation with RFX-family factors is an ancient, conserved phenomenon important for cilium specification. These results provide insight into a long-standing question about factors required for the differentiation of ciliated neurons in *C. elegans* and provide a prediction that a similar mechanism acts in sensory cilia in vertebrates. However, some concerns about interpretation of the data call into question whether it is legitimate to think of FKH-8 as a master regulator that acts syndergistically with DAF-19.

– Figure 6: it's nice to see (in the Methods) that a formal quantitative approach is used to test synergy between daf-19 and fkh-8. I'm concerned, however, that in this analysis the computed "expected" value assuming no synergy seems to have no associated error. This will make the one-sample t-test less stringent than it should be – that is, it will be easier for the experimentally observed value to appear significantly different from the expected value if there is no error associated with the expected value. I'm not sure there's any easy way around this, but the authors should provide stonger evidence that fkh-8 and daf-19 function synergistically.

– Authors argue that tm292 is a hypomorph based on its molecular structure and less severe phenotypes compared to vlc43. However, the truncation makes it seem possible that it could act as a dominant-negative/antimorphic allele. If so, this might affect the conclusions about synergy made in the double-mutant experiment, as nearly all of the experiments are carried out with tm292. Asking whether tm292 phenotypes are recessive would help to resolve this.

*Reviewer #2 (Recommendations for the authors):*

The transcriptional mechanisms driving the ciliogenic program are incompletely understood. Members of the RFX and Forkhead domain family of TFs have been implicated in motile ciliogenesis, and RFX TFs also regulate the expression of ciliogenic genes and regulate the development of primary cilia. Here the authors propose that the FKH-8 Forkhead domain TF works with the DAF-19 RFX TF in the development of primary cilia in *C. elegans*. Strengths of this work include the extensive bioinformatics and experimental analysis, and the finding that a subset of the fkh-8 mutant phenotypes can be rescued by their vertebrate homologs. However, the role of FKH-8 as a master regulator of ciliogenesis is not well-supported by the presented data. In particular, exactly how and where this TF acts in relation to the DAF-19 RFX remains unclear.

1. There are a few cases where an observation is reported but then not further followed up or discussed. This makes for a disjointed narrative. Examples:

Line 160: The authors report that mutations in daf-12 also have an effect on ciliary gene expression but this is not addressed any further. Given this, all data shown with daf-19; daf-12 should include not just WT as controls, but also the daf-12 mutant background as the directly relevant control (Figure 1C for example and others). Also see point 8 below.

Line 195: Figure 1E is another example. The authors report the enrichment of two PWMs in the upstream regions of genes containing X-boxes. But again, these are not followed up experimentally at all. Are these relevant in any way?

2. In Figure 1C and elsewhere, the authors report the overall number of cells expressing specific ciliary proteins. However, in addition to number, the level of expression is also a critical parameter. This is not described. A particular problem here is that all ciliary gene expression is examined using extrachromosomal reporters and the mosaicism of expression from these reporters together with their overexpression make precise measurements challenging. The authors need to validate at least a subset of their claims regarding effects of fkh-8, daf-19 and double/triple mutations on ciliary gene expression by examining the expression of GFP knockins at the relevant endogenous loci. Mutations in predicted FKH-8 binding sites should also be made at at least a subset of the endogenous loci and not only in these reporter constructs.

3. A particularly interesting category is the set of ciliary genes (broad and/or subtype-specific) that do NOT contain a predicted X-box. Is there any indication that these are regulated by any other shared motif?

4. It really wasn't clear to me from the presented analysis, exactly what fraction of ciliary genes containing X boxes also contain a predicted FKH-8 binding site and vice versa. Also, there is no clear data presented regarding the location and number of predicted FKH-8 binding sites relative to the X-boxes across the dataset (except for the few from the ChIP-seq dataset). These analyses need to be clearly presented.

5. The cilia morphology defects in fkh-8 mutants are very minor especially as compared to the drastic phenotypes of daf-19 mutants. It is difficult to assess how the shown effects on the expression of a subset of ciliary genes correlates with the shown ciliary morphological phenotypes. Did the authors look at cilia morphologies in the double/triple? How do they account for lengthening of one cilia type and shortening of others? I noted that cilia imaging was performed using sodium azide – this reagent shortens cilia. These measurements should be redone using levamisole.

6. The biggest issue with this work is that it is unclear whether the observed synergism is arising as a consequence of partly redundant effects on gene expression in individual neurons or across the population or both. In other words, there are at least two possible models. In the first, FKH-8 and DAF-19 could be acting together to regulate say ift-20 gene expression in every ciliated cell with perhaps somewhat each TF having greater or lesser contribution in individual cell types. In the second, these proteins could be acting to regulate ift-20 in different cell types. Either scenario would lead to an observed synergism in gene expression defects as measured in a population. Thus the statement in lines 521-22 regarding the mechanism of the synergism is not supported by the data.

What the authors need to do is to select a small number of genes, create endogenous reporter knock-ins (maybe consider using split-gfp to make it easier to look at single neurons) and then analyze gene expression levels in detail in a few defined ciliated sensory neurons in the singles and double/triples. The population level analysis is a good start but does not provide the resolution needed to really understand the underlying regulatory principles.

7. The forkhead domain protein FKH-2 was previously also implicated in regulation of expression of ciliary genes specifically in the AWB olfactory neurons and proposed to work together with DAF-19 (PMID: 17510633). This work not only needs to be referenced and discussed but perhaps also examined further in the context of the current work (especially since the authors also observe defects in AWB cilia).

8. Figure 6D examines expression of a subset of ciliome genes in fkh-8; daf-19; daf-12 triple mutants. It is unclear from the figure whether expression of ciliome genes in the triple mutant is significantly different from that of the daf-19; daf-12 double mutant. This is particularly critical because these are the only data addressing whether FKH-8 and DAF-19 act synergistically.

9. The rescue of the ift-20 expression phenotype by the vertebrate homologs is nice, but this result would be more convincing if additional phenotypes of fkh-8 mutants were also assessed for rescue.

---

## [Author Response]

[Editors’ note: the authors resubmitted a revised version of the paper for consideration. What follows is the authors’ response to the first round of review.]

Reviewer #1 (Recommendations for the authors):In this paper, Brocal-Ruiz et al. set out to identify factors that work together with the RFX factor DAF-19 in the development of sensory cilia in *C. elegans*. Using an informatic approach, they identify the Forkhead-family transcription factor FKH-8 as a candidate for this role. Their findings largely support the idea that fkh-8 is expressed broadly in sensory cilia, that it acts cell-autonomously and directly to activate the expression of many cilium-specific genes, and that full expression of these genes may depend on synergy between DAF-19 and FKH-8. Consistent with this, animals lacking fkh-8 have defects in cilium morphology and in many sensory behaviors that require the function of ciliated neurons. Interestingly, they provide evidence that some vertebrate Forkhead-family factors can substitute for fkh-8 in *C. elegans*, suggesting that cooperation with RFX-family factors is an ancient, conserved phenomenon important for cilium specification. These results provide insight into a long-standing question about factors required for the differentiation of ciliated neurons in *C. elegans* and provide a prediction that a similar mechanism acts in sensory cilia in vertebrates. However, some concerns about interpretation of the data call into question whether it is legitimate to think of FKH-8 as a master regulator that acts syndergistically with DAF-19.– Figure 6: it's nice to see (in the Methods) that a formal quantitative approach is used to test synergy between daf-19 and fkh-8. I'm concerned, however, that in this analysis the computed "expected" value assuming no synergy seems to have no associated error. This will make the one-sample t-test less stringent than it should be – that is, it will be easier for the experimentally observed value to appear significantly different from the expected value if there is no error associated with the expected value. I'm not sure there's any easy way around this, but the authors should provide stonger evidence that fkh-8 and daf-19 function synergistically.

We thank both reviewers for pointing out the importance of further addressing the synergistic actions of FKH-8 and DAF-19. Regarding the specific issue of the error associated to expected values we have now used propagation error to assign an error to the expected values (Supplementary File 1). This error is calculated as the sum of each standard deviation between the corresponding mean from each observed value in each double mutant multiplied by the value of the expected value. It is important to point out however that this has no impact on the statistical test used for the analysis of synergistic effects.

To provide stronger evidence for synergistic effects between these two transcription factors we have followed Reviewer #2 suggestion and focus on specific subpopulations of sensory neurons to quantify the phenotypes in the different genetic backgrounds. We show data for 3 different neuron types (CEPV, CEPD and PHA) and two different ciliome gene reporters (*osm-1* and *xbx-1*). These are examples in which reporter's expression is not affected or only very partially affected in each single mutant but completely absent from the double mutant. These constitute important results strongly supporting synergistic effects between FKH-8 and DAF-19. These data are now presented in Figure 4E.

– Authors argue that tm292 is a hypomorph based on its molecular structure and less severe phenotypes compared to vlc43. However, the truncation makes it seem possible that it could act as a dominant-negative/antimorphic allele. If so, this might affect the conclusions about synergy made in the double-mutant experiment, as nearly all of the experiments are carried out with tm292. Asking whether tm292 phenotypes are recessive would help to resolve this.

We appreciate raising this concern, we have now included data on the recessive nature of both the *tm292* and *vlc43* alleles (Figure S6C).

Reviewer #2 (Recommendations for the authors):The transcriptional mechanisms driving the ciliogenic program are incompletely understood. Members of the RFX and Forkhead domain family of TFs have been implicated in motile ciliogenesis, and RFX TFs also regulate the expression of ciliogenic genes and regulate the development of primary cilia. Here the authors propose that the FKH-8 Forkhead domain TF works with the DAF-19 RFX TF in the development of primary cilia in *C. elegans*. Strengths of this work include the extensive bioinformatics and experimental analysis, and the finding that a subset of the fkh-8 mutant phenotypes can be rescued by their vertebrate homologs. However, the role of FKH-8 as a master regulator of ciliogenesis is not well-supported by the presented data. In particular, exactly how and where this TF acts in relation to the DAF-19 RFX remains unclear.1. There are a few cases where an observation is reported but then not further followed up or discussed. This makes for a disjointed narrative. Examples:Line 160: The authors report that mutations in daf-12 also have an effect on ciliary gene expression but this is not addressed any further. Given this, all data shown with daf-19; daf-12 should include not just WT as controls, but also the daf-12 mutant background as the directly relevant control (Figure 1C for example and others). Also see point 8 below.

We thank the reviewer for pointing this out and apologise for the disjointed narrative. Regarding the specific issue about *daf-12*, and considering the focus of the paper is FKH-8 characterization, we have decided to follow the same strategy as previous works using *daf-12(sa204); daf-19(m86)* mutants in which *daf-12(sa204)* is always used as the relevant control (*doi:10.1016/j.ydbio.2011.06.028; doi:10.1534/genetics.117.300571/-/DC1.1)*.

Thus, in the new manuscript whenever *daf-12(sa204); daf-19(m86)* is used the control refers to *daf-12(sa204)* (Figure 1 and Figure 4). To avoid disjointed narrative we have omitted the comment on *daf-12* effect on ciliary gene expression.

Line 195: Figure 1E is another example. The authors report the enrichment of two PWMs in the upstream regions of genes containing X-boxes. But again, these are not followed up experimentally at all. Are these relevant in any way?

We thank again the reviewer for helping improve the flow of the manuscript. We found that the two additional motifs found in genes containing X-boxes show strong similarity to DAF-19 and to the previously reported C-BOX (*doi:10.1016/j.ydbio.2012.05.033)*. Thus, these motifs are not useful to identify new factors involved in the regulation of ciliogenesis. We have added this information in Line 200-202 and move the panel to Figure S3C to help the reader stay focused on the main message of the paper.

2. In Figure 1C and elsewhere, the authors report the overall number of cells expressing specific ciliary proteins. However, in addition to number, the level of expression is also a critical parameter. This is not described. A particular problem here is that all ciliary gene expression is examined using extrachromosomal reporters and the mosaicism of expression from these reporters together with their overexpression make precise measurements challenging. The authors need to validate at least a subset of their claims regarding effects of fkh-8, daf-19 and double/triple mutations on ciliary gene expression by examining the expression of GFP knockins at the relevant endogenous loci. Mutations in predicted FKH-8 binding sites should also be made at at least a subset of the endogenous loci and not only in these reporter constructs.

We appreciate reviewer's comments on the use of extrachromosomal arrays. Extrachromosomal multicopy arrays have been extensively used in the past to provide robust evidence for the relevant targets for many transcription factors. Undoubtedly, mosaicism associated to Ex arrays is one caveat for their use, to avoid this issue, the same extrachromosomal line is crossed in the different genetic backgrounds to be able to compare and assess the effects associated to the mutations and not to the mosaicism. Nevertheless, we understand CRISPR endogenous tagging has become the gold standard in the field, thus we built an endogenously tagged *osm-5* strain (*osm-5::SL2::GFP::H2B* ) as a proof of concept to assess the role of FKH-8 in core ciliome gene expression. Our results show that *fkh-8(vlc43)* displays broad defects in *osm5* endogenous expression. Following reviewer #2 advice, we have quantified fluorescence levels in specific subpopulations of sensory neurons. These data are now presented in Figure 3 C,D.

3. A particularly interesting category is the set of ciliary genes (broad and/or subtype-specific) that do NOT contain a predicted X-box. Is there any indication that these are regulated by any other shared motif?

There are 61 ciliome genes that do not show an X-BOX in the analysed regulatory sequences. Most of them (43 genes) correspond to subtype specific ciliome components. We did not find enriched motifs for known transcription factors (this is now included in the text line 203-209). It is important to mention that many subtype specific ciliome components are known to be directly regulated by the terminal selector acting in each specific sensory ciliated neuron (doi:10.1038/nature07929, doi: 10.1101/gad.1560107; doi:10.1242/dev.099721). Failure to identify common motifs in subtype specific ciliome components might be reflecting this subtype specific regulation by terminal selectors. This point has been included in the discussion (Line 608611)

4. It really wasn't clear to me from the presented analysis, exactly what fraction of ciliary genes containing X boxes also contain a predicted FKH-8 binding site and vice versa. Also, there is no clear data presented regarding the location and number of predicted FKH-8 binding sites relative to the X-boxes across the dataset (except for the few from the ChIP-seq dataset). These analyses need to be clearly presented.

We thank very much the reviewer for this comment and apologise for the lack of clarity.

It is important to clarify that we use experimentally assessed FKH-8 ChIP-seq peaks and not predicted motifs, as the predicted binding site for FKH-8 has not been experimentally determined. Predicted motifs for FKH TFs (RYMAAYA for the classical FKH motif and GACGC for the FKH like) are small and degenerate and thus not very informative for predictions in the genome.

Regarding DAF-19, there is no available ChIP-seq data, however X-boxes correspond to long imperfect palindromic sequences with high information content (Figure S3A). They are so informative that their mere presence in the promoters of novel genes has been used to identify new ciliome components in different organisms including *C. elegans* (doi: 10.1186/gb-2006-7-12r126; doi: 10.1186/gb-2006-7-12-r126; doi:10.1016/j.cub.2005.04.059). Moreover, many of these predicted X-boxes have been previously verified by site directed mutagenesis. Finally, many bioinformatic hits of X-boxes compare extremely well to experimentally determined vertebrate RFX TF binding (DOI: 10.1016/j.celrep.2022.110661; doi.org/10.1038/35002634). Thus, most Xboxes are likely to correspond to DAF-19 bound sequences.

Our data shows that most core ciliome genes contain both an X-box and a FKH-8 peak, in addition, FKH-8 genomic binding takes place in close proximity to X-boxes (center of FKH-8 peak is less than 600 bp apart from the X-box motif).

Importantly, we complemented this information with coIP experiments that show DAF-19D and FKH-8 can physically interact.

All these new data are presented in Figure #2F-H and Figure S5.

5. The cilia morphology defects in fkh-8 mutants are very minor especially as compared to the drastic phenotypes of daf-19 mutants. It is difficult to assess how the shown effects on the expression of a subset of ciliary genes correlates with the shown ciliary morphological phenotypes. Did the authors look at cilia morphologies in the double/triple? How do they account for lengthening of one cilia type and shortening of others? I noted that cilia imaging was performed using sodium azide – this reagent shortens cilia. These measurements should be redone using levamisole.

We thank the reviewer for these comments. Several pieces of evidence indicate that *daf-19* mutants show stronger phenotypes: (1) Reporter gene expression is more affected in *daf-19* than in *fkh-8* mutants (Figure 1C compared to Figure 3A), (2) Lipophilic staining with DiI is affected in *daf-19* but not in *fkh-8* mutants and (3) There are smaller ciliary morphology defects in *fkh-8* compared to complete absence of cilia in *daf-19* mutants (doi:10.1016/0012-1606(86)90314-3) (4) Our new data shows that DAF-19 ectopic expression, but not FKH-8, is sufficient to drive ciliary gene expression (Figure S9). Accordingly, we have now omitted the term master regulator to refer to FKH-8 ciliogenic actions.

Following reviewer's suggestion, there are some links that can be extrapolated between *fkh-8* ciliome gene expression defects and morphology and behaviour defects. For example, our new data shows broad expression defects for *osm-5* endogenous gene reporter in *fkh-8* mutants (Figure 3C), and *osm-5* mutants show AWB cilia morphology defects (doi:10.1038/sj.emboj.7601717) which are also present in *fkh-8* mutants (Figure 5). Finally, repulsion to 2-nonanol, that is mediated by AWB, is also affected in *fkh-8* mutants (Figure 6). We now refer to these correlations of phenotypes directly in the text (Line 447 to 449 and 505 and 507).

Nevertheless, both ciliary morphology and behaviour are functional readouts of cilia integrity that is regulated by a great number of different genes, thus mutations in many ciliome components produce cilia morphology defects or behaviour defect. It is hard to assign specific targets to specific defects. The relevance of our manuscript is the description of the role of FKH-8 in the DIRECT regulation of a BROAD number of ciliome components, COOPERATING with DAF-19 (which we now show further support of synergistic actions and physical interaction). Thus, it is probable that *fkh-8* mutant defects in ciliary morphology and cilia regulated behaviours could be attributed to gene expression defects in several ciliome components.

We appreciate the suggestion of studying synergistic effects of DAF-19 and FKH-8 in cilia morphology however *daf-19* mutants lack cilia (doi:10.1016/0012-1606(86)90314-3), thus we cannot perform those experiments.

We thank the reviewer suggesting that we measure cilia length using levamisole. We found *fkh8* mutants are extremely resistant to levamisole, alternatively we used beads to immobilise worms and reproduced morphology defects in the mutant (Figure S10).

Finally, cilia length is controlled by a balance between cilia assembly and disassembly regulated by IFT and mutants for ciliome components can produce both shortened or elongated cilia. We have added this explanation in the text (Line 435-437). Nevertheless, we should be cautious with the lengthening phenotype in ADF in *fkh-8* mutants because when assessed in bead immobilized worms ADF cilia in *fkh-8* mutants is shorter than in *wild type*, this is now included in the text (Figure S10).

6. The biggest issue with this work is that it is unclear whether the observed synergism is arising as a consequence of partly redundant effects on gene expression in individual neurons or across the population or both. In other words, there are at least two possible models. In the first, FKH-8 and DAF-19 could be acting together to regulate say ift-20 gene expression in every ciliated cell with perhaps somewhat each TF having greater or lesser contribution in individual cell types. In the second, these proteins could be acting to regulate ift-20 in different cell types. Either scenario would lead to an observed synergism in gene expression defects as measured in a population. Thus the statement in lines 521-22 regarding the mechanism of the synergism is not supported by the data.What the authors need to do is to select a small number of genes, create endogenous reporter knock-ins (maybe consider using split-gfp to make it easier to look at single neurons) and then analyze gene expression levels in detail in a few defined ciliated sensory neurons in the singles and double/triples. The population level analysis is a good start but does not provide the resolution needed to really understand the underlying regulatory principles.

We are very grateful to the reviewer for this suggestion. Our initial data suggest the effects are higher than what will be expected by an additive effect, either at the cell or at the population level, nevertheless, we agree with the reviewer this population level analysis does not have enough resolution to strongly claim synergistic effects. An additional important limitation with the population analysis is the ectopic reporter expression observed for some ciliary gene reporters in *daf-19* mutants, which could mask missing expression in ciliated neurons in the different mutant backgrounds.

Thus, following reviewer's advice we aimed to identify specific subpopulations of ciliated neurons. First, we attempted to use Neuropal (DOI: 10.1016/j.cell.2020.12.012) to identify each different ciliated neuron type, however, although NeuroPAL worked well for neuron identification in the *wild type* background, NeuroPAL reporters were massively missregulated in *daf-19* mutants precluding its use to identify specific neuron types in double and triple mutants.

Alternatively, we first tentatively identified neuron-types candidates to display synergy based on population scores and cell locations and next selected specific reporter strains unaffected in all genetic backgrounds to unequivocally identify those neuron types. This was a laborious task but turned out to be very important. In this way we have been able to assess synergistic effects of DAF19 and FKH-8 in the regulation of *osm-1* reporter in CEPV and CEPD and of *xbx-1* reporter expression in PHA (Figure 4E).

We think this new set of experiments, together with additional data (co-expression of DAF-19 and FKH-8 in all ciliated neurons, binding in nearby regions of the regulatory genome of ciliome genes and physical interaction) provide strong evidence for synergistic actions for both TFs.

7. The forkhead domain protein FKH-2 was previously also implicated in regulation of expression of ciliary genes specifically in the AWB olfactory neurons and proposed to work together with DAF-19 (PMID: 17510633). This work not only needs to be referenced and discussed but perhaps also examined further in the context of the current work (especially since the authors also observe defects in AWB cilia).

We thank the reviewer for this comment and for the suggested experiments. We now cite this work and have now determined that *fkh-2* expression in AWB is unaffected in *fkh-8* null mutants. The data is now indicated in results (Line 374-386) and discussion (Line 594-596).

8. Figure 6D examines expression of a subset of ciliome genes in fkh-8; daf-19; daf-12 triple mutants. It is unclear from the figure whether expression of ciliome genes in the triple mutant is significantly different from that of the daf-19; daf-12 double mutant. This is particularly critical because these are the only data addressing whether FKH-8 and DAF-19 act synergistically.

Following reviewer #2 suggestion we have now addressed synergism analysing specific sensory ciliated neuron subtypes (Figure 4D).

We have nevertheless maintained total quantification of cells for *ift-20* (Figure 4C) and for *xbx-1* and *peli-1* reporters (Figure S9). For all three reporters, mean number of expressing neurons in *daf-12(sa204); daf-19(m86)*, *fkh-8(tm292)* triple mutants is significantly different from each of the double mutants and significantly lower than the expected from the multiplicative effect of both *daf-12(sa204); fkh-8(tm292)* and *daf-12(sa204); daf-19(m86)* animals. This information is now indicated in the corresponding figure legends.

9. The rescue of the ift-20 expression phenotype by the vertebrate homologs is nice, but this result would be more convincing if additional phenotypes of fkh-8 mutants were also assessed for rescue.

We agree with the reviewer rescuing of additional phenotypes with mouse orthologs will be nice, however these experiments will not provide any conceptually addition to the manuscript. We decided to prioritize the work on the synergistic actions of DAF-19 and FKH-8, for which we have now provided new data on double mutants and also physical interaction. In addition, as part of a new project in the laboratory, we are performing a systematic analysis on the rescuing ability of different members of the mammalian FKH family in *C. elegans*. In this context, we will also expand the panel of rescued reporters, but we think performing these experiments is beyond the scope of this manuscript and will further delay its publication.